# Selection for infectivity profiles in slow and fast epidemics, and the rise of SARS-CoV-2 variants

François Blanquart[1]*, Nathanaël Hozé[2], Benjamin John Cowling[3,4], Florence Débarre[5], Simon Cauchemez[2]

[1]Centre for Interdisciplinary Research in Biology (CIRB), Collège de France, CNRS INSERM, PSL Research University, Paris, France; [2]Mathematical Modelling of Infectious Diseases Unit, Institut Pasteur, Université de Paris, UMR2000, CNRS, Paris, France; [3]WHO Collaborating Centre for Infectious Disease Epidemiology and Control, School of Public Health, Li Ka Shing Faculty of Medicine, The University of Hong Kong Pokfulam, Hong Kong Special Administrative Region, Hong Kong, China; [4]Laboratory of Data Discovery for Health, Hong Kong Science and Technology Park New Territories, Hong Kong Special Administrative Region, Hong Kong, China; [5]Institute of Ecology and Environmental Sciences of Paris (iEES-Paris, UMR 7618) CNRS, Sorbonne Université, UPEC, IRD, INRAE, Paris, France

*For correspondence: francois.blanquart@college-de-france.fr

**Competing interest:** The authors declare that no competing interests exist.

**Abstract** Evaluating the characteristics of emerging SARS-CoV-2 variants of concern is essential to inform pandemic risk assessment. A variant may grow faster if it produces a larger number of secondary infections ("R advantage") or if the timing of secondary infections (generation time) is better. So far, assessments have largely focused on deriving the R advantage assuming the generation time was unchanged. Yet, knowledge of both is needed to anticipate the impact. Here, we develop an analytical framework to investigate the contribution of *both* the R advantage and generation time to the growth advantage of a variant. It is known that selection on a variant with larger R increases with levels of transmission in the community. We additionally show that variants conferring earlier transmission are more strongly favored when the historical strains have fast epidemic growth, while variants conferring later transmission are more strongly favored when historical strains have slow or negative growth. We develop these conceptual insights into a new statistical framework to infer both the R advantage and generation time of a variant. On simulated data, our framework correctly estimates both parameters when it covers time periods characterized by different epidemiological contexts. Applied to data for the Alpha and Delta variants in England and in Europe, we find that Alpha confers a+54% [95% CI, 45–63%] R advantage compared to previous strains, and Delta +140% [98–182%] compared to Alpha, and mean generation times are similar to historical strains for both variants. This work helps interpret variant frequency dynamics and will strengthen risk assessment for future variants of concern.

## Editor's evaluation

This manuscript will be of broad interest to readers interested in understanding the characteristics of variants in ongoing epidemics that lead to faster (or slower) growth rates and will be of particular interest to those wishing to understand the factors leading to the selection of SARS–CoV–2 variants. The selective advantage of a novel strain of a pathogen depends not only on its relative transmissibility but also on its generation time relative to other strains; the relation between transmissibility, transmission advantage and generation time changes across different phases of the epidemic. Key

innovations in this paper are a robust framework for using this relationship to make statistical inferences about both the transmissibility advantage and generation time of an emerging variant and conceptual novelty in the general investigation of selection on infectivity profiles. The approach is supported by simulation studies and applied to the Alpha and Delta SARS–CoV–2 variants to show that selection was likely driven by changes in transmissibility rather than changes in the generation time.

## Introduction

Human pathogens rapidly adapt to their hosts. During the short evolutionary history of SARS-CoV-2 with humans, since the host shift in late 2019, several selected mutations and combinations of mutations (variants) have emerged: for example, the D614G mutation in Spring 2020 (*Volz et al., 2021a*), the variant Alpha in Fall 2020 (B.1.1.7), and Delta in Spring 2020 (B.1.617.2). Variant Alpha, first detected in England in September 2020, rapidly rose in frequency in October 2020 in England and spread to multiple countries, causing important rebounds of the epidemic (*Volz et al., 2021b*; *Davies et al., 2021*; *Borges et al., 2021*; *Washington et al., 2021*; *Gaymard et al., 2021*). Later on, the Delta variant became dominant in many parts of the world, leading to new surges in infections and hospitalizations in 2021.

Each time a variant of concern emerges, it is essential to determine whether it has acquired a competitive advantage compared to circulating SARS-CoV-2 strains, in order to anticipate the potential effects on the epidemic trajectory and hospital admissions. The advantage can be caused by an increased capacity to transmit or to escape the immune response acquired by previous infection or vaccination. The transmission advantage, hereafter *R advantage,* of an emerging variant has been defined as the relative increase in the effective reproduction number *R* (average number of secondary cases). It has often been estimated by analysing the rise in the variant's frequency, under the assumption that all strains shared the same generation time distribution (i.e. delay between infection in the primary case and the people they infect). However, this has sometimes led to unstable estimates of the R advantage. For example, in the United Kingdom, the R advantage of the Alpha variant was estimated to decline from +89 to +54% from December to mid-January. Estimates of R advantage of Alpha and Delta across countries also exhibit substantial variability (*Campbell et al., 2021*). The reasons why estimates of the R advantage would vary with the epidemiological context remain obscure. The instability this generates is problematic for planning since it affects medium and long-term evaluations of the epidemic trajectory. It also suggests that methods currently used to ascertain the risks posed by emerging variants may miss important drivers of emergence.

The evolutionary fate of an emerging variant is ultimately determined by its exponential growth rate relative to that of the circulating strains. When evaluating the growth advantage of a variant, the focus has so far largely been on estimating differences in *R*. However, the growth rate of an emerging variant depends not only on the effective reproduction number but also on the generation time distribution. Natural selection is therefore expected to act on the full infectivity profile, which combines both *R* and the generation time. Previous theoretical work has explored how selection acts on various pathogen life-history traits (*Lenski and May, 1994*; *Day and Proulx, 2004*; *Day and Gandon, 2007*; *Day et al., 2020*), with most often a focus on selection for transmissibility and virulence (disease-induced mortality). Recent work on SARS-CoV-2 mentioned the possibility of selection on the time interval between infection and infectiousness, but did not characterize the sign and magnitude of this component of selection in detail (*Day et al., 2020*).

Here, we develop a mathematical model to investigate how the growth advantage may vary with the epidemiological context when variants have different generation times. We use the relationship between infectivity profile and growth rate to explore conceptually how infectivity profiles are selected for. We find that the growth advantage of a variant generally depends on the epidemiological context, more precisely on whether the transmission is low (declining or slow epidemic) or high (fast epidemic). We call "transmission" the rate of infectious contacts in the community, which depends on the epidemiological context, for example, social distancing, mask-wearing, population immunity, seasonal effects, etc. We use the time-varying reproduction number of the historical strain (taken as a reference), denoted $R_H(t)$, as a proxy for transmission. Our conceptual exploration suggests that it is possible to infer changes in infectivity profile associated with an emerging variant, and not only

**eLife digest** Mutations in genes of the SARS-CoV-2 virus have generated new variants of concern, like Alpha, Delta, and more recently Omicron. These strains contain genetic modifications that help the virus spread more easily as well as altering the severity of the illness it causes. This has led to rising numbers of infections, known as epidemic waves, in many parts of the world.

Tracking new variants of concern is crucial to protecting the public. To do this, scientists monitor how many people one person with the virus can infect, also known as the **number of secondary infections.** They may also measure when in the course of the illness an individual may pass along the virus to others. Together, these metrics help determine how fast and large an outbreak caused by a new variant will grow. The more people the new variant infects and the quicker it spreads, the more likely it is to replace existing strains of the virus.

So far, most studies have assumed that the growth rate of a new variant solely depends on the number of secondary infections, and the timing of secondary infections is often not considered. To address this, Blanquart et al. built a mathematical model that combines both these parameters to determine the growth rate of new viral strains.

The model showed that variants which rapidly cause secondary infections have a larger growth advantage over existing strains when the virus is more easily transmitted between individuals and the epidemic spreads rapidly. But when there is less transmission and the epidemic is declining, variants that generate secondary infections after a longer time have an advantage. For example, when control measures like mask wearing or social distancing are in place, delayed secondary infections may be more advantageous.

Blanquart et al. then applied their model to data from the Alpha and Delta variant outbreaks in the United Kingdom. They found that Alpha and Delta did not change the timing of secondary infections compared to previously circulating strains. But the Alpha variant had a 54% transmission advantage over previous strains and the Delta variant had a 140% transmission advantage over Alpha.

Taken together, these findings suggest that the timing of secondary infections and transmission rates both play an important role in how quickly a virus spreads. The new mathematical model created by Blanquart et al. may help epidemiologists better predict the trajectory of new SARS-CoV-2 variants and determine how to best control their spread.

its growth rate, in settings where transmission changes over time. We thus develop an inference framework to estimate changes in infectivity profile associated with a variant and more precisely characterize the R advantage of the variant and its dependency to the epidemiological context. We apply our approach to data on Alpha and Delta variants frequency over time in England in Fall 2020 and Spring 2021. On the methodological front, the originality of our work lies in the use of the Euler-Lotka equation to explore generally how infectivity profiles are selected for (in contrast to previous work using ordinary differential equation models), and in the development of an inference framework.

## Results
### General Principle
We consider a situation where a variant, or a set of strains with similar properties, has been circulating for some time (historical strains *H*) and is challenged by an emerging variant *E*. When investigating the rise of variant *E* relative to historical strains *H*, we typically only observe the exponential growth rates of *H* and *E* strains, which translates into a growth advantage (or disadvantage). More precisely, if the historical strains grow with an exponential rate $r_H$ and the emerging variant grows with rate $r_E$, the logit of the frequency of the emerging variant will grow linearly with slope $r_E - r_H$ . This slope is called the growth advantage or selection coefficient. The selection coefficient of the emerging variant relative to historical strains is obtained by a linear regression of the logit frequency of the variant over time. It is then typically assumed that both variant and historical strains share the same distribution of generation time. The effective reproduction number of the emerging variant relative to historical strains, called the R advantage ($R_E/R_H - 1$), can thus be deduced.

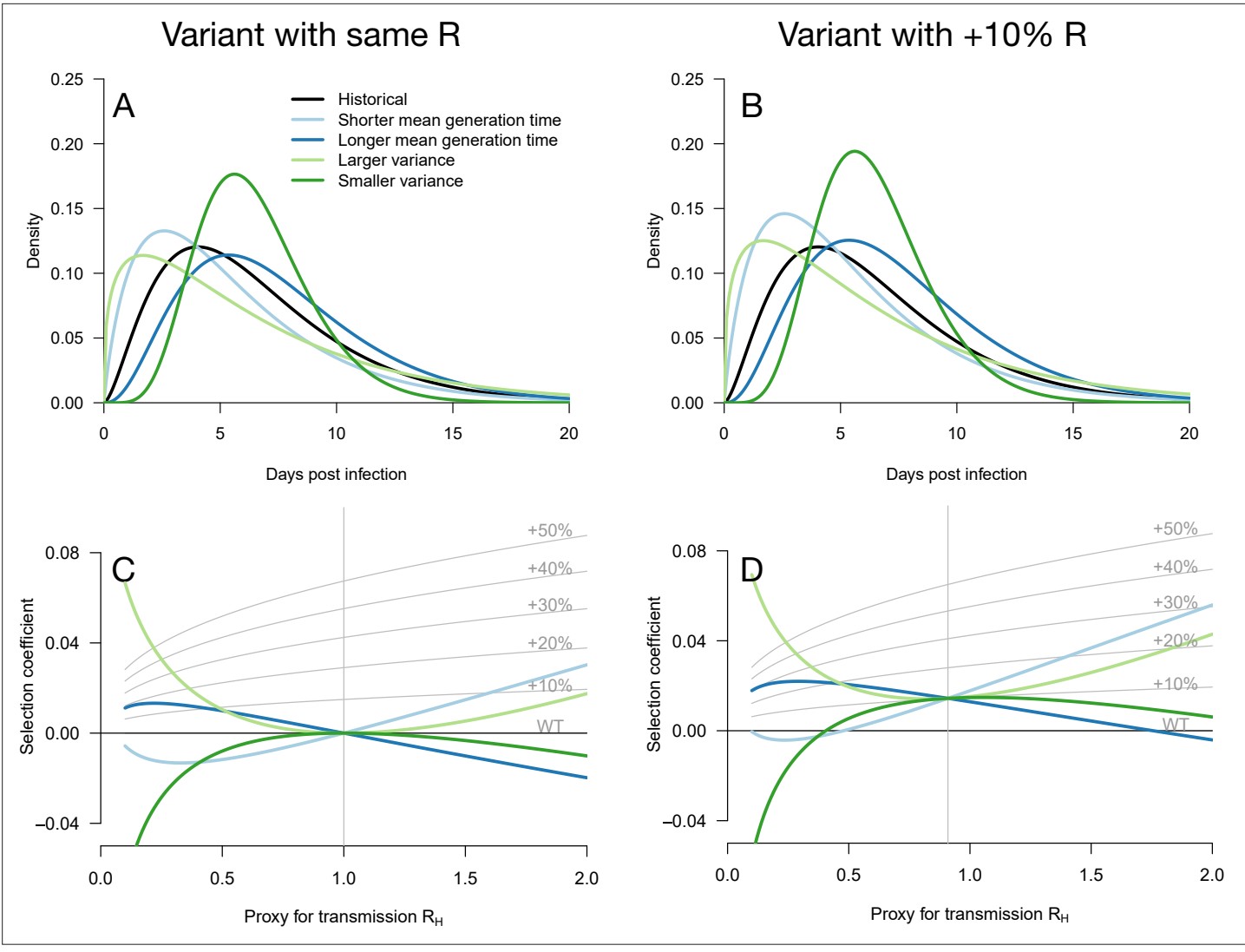

**Figure 1.** Variation of the selection coefficient as a function of transmission, for several infectivity profiles of emerging variants. The Panels A and B show several variant infectivity profiles with the same effective reproduction number as historical strains (**A**) or an effective reproduction number increased by +10% (**B**). For variants with shorter or longer mean generation time, the relative mean generation time is –15 or +15% compared to historical strains. For variants with shorter or longer standard deviation (sd) in generation time, the relative sd in generation time is –40 or +40% compared to historical strains. The Panels C and D show the selection coefficient of these variants as a function of the proxy for transmission (effective reproduction number of the historical strains). The gray lines show the selection coefficient if the variant had the same infectivity profile as historical strains, with the infectivity profile increased uniformly by a constant as commonly assumed.

Yet natural selection acts on the full infectivity profile of a strain, not only on the effective reproduction number. The infectivity profile is a function $\beta\left(\tau\right) = R\,w\left(\tau\right)$ of the time since infection $\tau$, where $R$ is the effective reproduction number, while $w\left(\tau\right)$ characterizes the distribution of generation time and satisfies $\int_{\tau=0}^{\infty} w\left(\tau\right) d\tau = 1$.

## Selection for Infectivity Profiles in Slow and Fast Epidemics

We first examine how the selection coefficient $r_E - r_H$ varies with transmission for various infectivity profiles of the variant compared to historical strains (*Figure 1*). To do so, we use the Euler-Lotka equation, relating the growth rate to the effective reproduction number and the distribution of the generation time (Methods). This equation assumes that each variant grows or declines exponentially and that the distribution of the time since infection is at any time at its equilibrium value: an exponential distribution with rate equal to the exponential growth rate (*Wallinga and Lipsitch, 2007*). This equilibrium distribution naturally follows from the assumption of exponential growth, which implies

that individuals infected $t + \Delta t$ days ago are $e^{r \, \Delta t}$ less numerous than individuals infected $t$ days ago, with $r$ the exponential growth rate.

We first note that even for variants with an R advantage but *not* affecting the distribution of generation time, the selection coefficient increases with transmission (*Day and Proulx, 2004*; *Day and Gandon, 2007*; *Day et al., 2020*). This relationship is shown in *Figure 1* as gray lines for a series of variants conferring a R advantage of +10 −+50% but not affecting the gamma-distributed generation time.

Several more general infectivity profiles of the emerging variant are now considered. We first consider variants with the same effective reproduction number as historical strains, which can be selected for if secondary infections occur at different timings (*Figure 1A and C*). A variant with a shorter mean generation time is selected for in a growing epidemic because it produces the same number of secondary infections in a shorter time, while the same variant is counter-selected in a declining epidemic (*Figure 1C*). The opposite holds for variants with a longer mean generation time. A variant with a larger standard deviation (sd) in generation time is always selected for because it enjoys both an excess of early secondary infections and late secondary infections compared to historical strains. The opposite holds for a variant with smaller sd in generation time. Generally, in a growing epidemic, more individuals have been infected recently and there is a greater advantage to early transmission, while in a declining epidemic, more individuals have late time since infection and therefore strains that transmit at later times have a greater advantage. Again, this is a direct demographic consequence of growth or decline of the epidemic. Selected variants could thus enjoy a more favorable timing of secondary infections, even in the absence of R advantage (*Park et al., 2021*).

Additionally, an advantage in the timing of secondary infections can of course be combined with a R advantage. We, therefore, consider hypothetical emerging variants with a range of generation time distributions and a+10% R advantage (*Figure 1B and D*). These variants generally have a stronger growth advantage, but the variation in selection coefficient with the transmission is similar to that of the analogous variants without the R advantage (*Figure 1C and D*).

Notably, in both cases, there is a level of transmission (expressed in terms of the effective reproduction number of the historical strain $R_H$) at which all variants perform similarly regardless of their distribution of generation time. This level of transmission is precisely where the size of the variant epidemic is constant in time (variant growth rate is 0). For example, this level of transmission is $R_H = 1$ for a variant not affecting R (*Figure 1C*). For a 10% R advantage, this level is $R_H = 1/1.1$ (*Figure 1D*). Generally, for a variant conferring a R advantage $\delta_1$, the level of transmission is $R_H = 1/\left(1 + \delta_1\right)$. At this level of transmission, the distribution of time since infection is uniform—all ages of infection are equally represented—and therefore the infectivity profile does not matter for the selection of the variant.

While these results assume that the time since infection is fixed at its equilibrium distribution, it is not necessarily the case in a realistic scenario where transmission changes over time. We applied these analytical results to simulations where the time since infection structure emerges from the model dynamics. The epidemiological dynamics are simulated with discrete-time renewal equations including time-varying transmission, the build-up of population immunity (assumed to be identical for historical strains and emerging variants), and detection and isolation of cases (Methods). In the context of progressive decline of levels of transmission, variants with the same reproduction number but distinct generation time distributions can exhibit a variety of epidemiological and frequency dynamics (*Figure 2A and C*). The epidemiological dynamics, that is the daily number of variant cases, are also strongly affected by the generation time distribution of the variant (*Figure 2A*). Frequency trajectories can be non-monotonous: a variant with shorter mean generation time could be selected, increase in frequency, but then decline in frequency as transmission declines (*Figure 2C*). Several variants all sharing the same 10% R advantage exhibit a range of frequency trajectories in the simulations: the trajectory of one of them initially resembles that of a variant with a+30% advantage and same generation time distribution as historical strains (*Figure 2D*, light blue line compared to +30% curve), while others stagnate at low frequency (*Figure 2D*, blue and green lines).

## Inference of the Infectivity Profiles of Variants

The variety of variant trajectories shown in *Figure 2* suggests that it may be possible to infer the changes in infectivity profiles of variants by analyzing their growth advantage as a function of

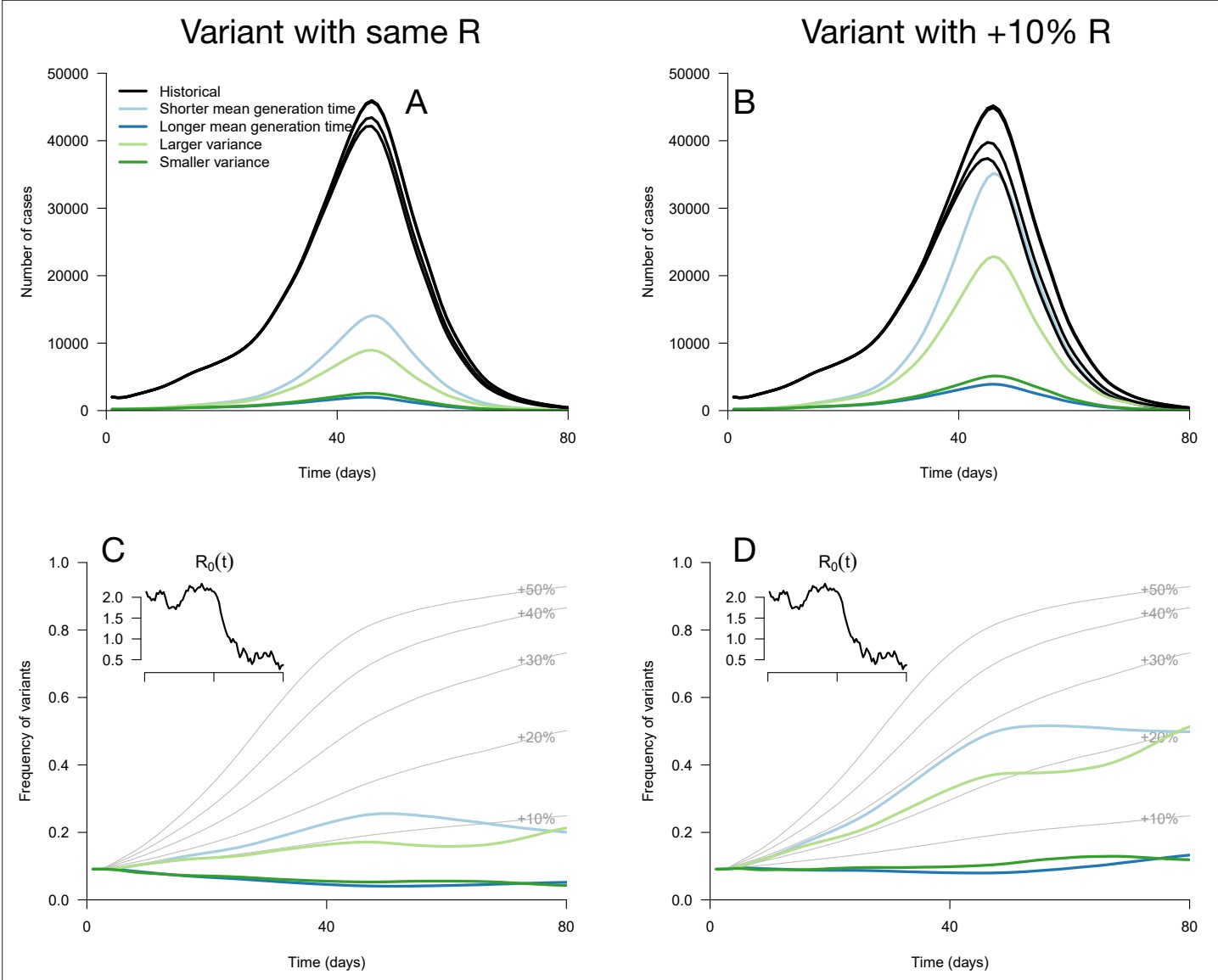

**Figure 2.** Epidemiological and evolutionary trajectories of several types of emerging variants competing with historical strains. The Panels A and B show epidemiological dynamics (daily cases number of historical strains in black, of variants in colours). Historical strains display several slightly different curves when competing with each of the variants because of the weak competition brought about by the build-up of population immunity. The Panels C and D show evolutionary trajectories, with daily variant frequency as colored lines. The light gray curves show the frequency dynamics of variants with a +10 to +50%R advantage, with the same generation time as historical strains. The inset shows the transmission through time in these simulations, given by the basic reproduction number of historical strains (see details of simulation model in Methods).

The online version of this article includes the following figure supplement(s) for figure 2:

**Figure supplement 1.** The relationship between estimated growth rate of the variant and historical strains across time in the simulation study, in the scenario of progressive decline of $R_{0,H}(t)$ from 1.5 to 0.5.

transmission when we have accumulated sufficient data documenting growth rates in different epidemiological situations. We first tested this idea on simulated data, in scenarios of emergence and progressive decline in transmission.

Our inference framework is based on the relationship between growth rates of historical strains ($r_H$) and of the emerging variant ($r_E$). Our aim is to infer three parameters $\delta_1$, $\delta_2$, $\delta_3$, characterizing respectively the R advantage, the mean generation time and the sd of generation time of the emerging strain relative to that of historical strains. The relationship can be used to infer properties of the variant infectivity profile, but not with a classical linear regression because both $r_H$ and $r_E$ are measured with

errors with a complex covariance structure. Furthermore, large errors can blur the subtle distinction between different profiles (*Figure 2—figure supplement 1*). The error variance is inversely related to the number of cases and the size of the sample used to assay variant frequency ("sample size"). When the total number of cases and the sample size are both large, the joint distribution of the time series of estimated $r_H$ and $r_E$ is approximately multivariate normal (Methods). The mean vector of the multivariate normal depends on the parameters $R_H(t)$ (the time series of effective reproduction number), on the mean and sd of the historical strains' generation time (fixed to 6.5 and 4 days respectively), and on the parameters of interest $\delta_1$, $\delta_2$, $\delta_3$. The variance-covariance structure of the multivariate normal distribution is fixed and depends on the daily total number of cases and the sample size. It is also possible to derive the multivariate normal distribution of $r_H$ and $r_E$ across different independent regions instead of over time (Methods).

With this statistical framework at hand, we conducted a simulation study to infer jointly the R advantage of the variants and their relative mean generation time (*Figure 3*). In these simulations, we systematically varied the infectivity profile of the emerging strain: R advantage ($\delta_1$) from +0 to +50% and relative mean generation time ($\delta_2$) from –40 to +40%. We assumed the emerging strain had the same sd of generation time as historical strains ($\delta_3 = 0$). We jointly inferred the two parameters $\delta_1$ and $\delta_2$ for different sample sizes (sample size, N=1000, 3000, 10,000 per day) and for different scenarios of variability in epidemiological context, from a strong to a weak decline. In the explicit simulations, the parameter that we tune is the *basic* reproductive number $R_{0,H}(t)$ assumed to decline from 1.5 to 0.5, 1.3 to 0.7, and 1.1 to 0.9 over 80 days in the three scenarios, and the inferred parameters represent the *effective* reproductive number after accounting for population immunity and case detection and isolation (*Figure 3—figure supplement 1*).

The R advantage was almost always precisely inferred even with small sample size and small variability in effective reproduction number (*Figure 3*, left panels). In contrast, the relative mean generation time was well inferred only for a large sample size and large variability in effective reproduction number (*Figure 3*, right panel). This was because the large error on $r_E$ and $r_H$ due to the limited sample size, in conjunction with the small variability in $R_{0,H}(t)$, makes precise inference of the relationship between $r_E$ and $r_H$ difficult. For such small sample sizes, grouping data by week (instead of by days, for the same duration) improves inference (*Figure 3*, green points).

We tested the robustness of the inference framework in several ways. First, we investigated whether temporal lag between infections and recorded cases impacted inference, considering the additional scenario where the mean time from symptom onset to case detection was 6 days instead of 2.2 days (*Figure 3—figure supplement 2*). The lag does not affect the accuracy of inference (compare with *Figure 3*), as expected since the time series of number of detected cases and variant frequency share the same lag with infections. Second, we applied our framework to additional simulations where the true generation time is not gamma-distributed as in our baseline model: we assumed that transmission is not possible during the first two days; this is followed by a shifted gamma-distributed timing of secondary infections. This mismatch between modeling assumption and the true generation time distribution led to a small overestimation of the relative mean generation time (*Figure 3—figure supplement 2*). Lastly, we inferred parameters when there is a sharp decline in $R_H(t)$ from 1.5 to 0.5 at a fixed date, instead of a progressive decline. In this scenario, the distribution of time since infection may not immediately stabilize after the sharp decline, temporarily breaking the key assumption of our analytical approach. Parameters were correctly inferred when the time series after the decline was long enough. With only 10 days of data after the decline, the inferred relative mean generation time was accurately inferred if enough cases were assayed to measure the variant frequency (*Figure 3—figure supplement 3*). Interestingly, accuracy was better for variants with a shorter generation time than the historical strains (negative relative mean generation time). This is probably because their distribution of time since infection stabilizes faster after the sharp decline.

## Application to the Variants Alpha and Delta in England

We used this framework to infer the R advantage and mean generation time of Alpha (B.1.1.7) in England. We used public data from Public Health England on weekly cases numbers and the frequency of the Alpha variant (dataset 1). In England, the frequency of the Alpha variant could be quickly inferred because some of the PCR tests for SARS-CoV-2 detection presented a "S gene target failure" (SGTF) caused by the spike 69/70 deletion characteristic of Alpha (*Public Health England, 2021b*).

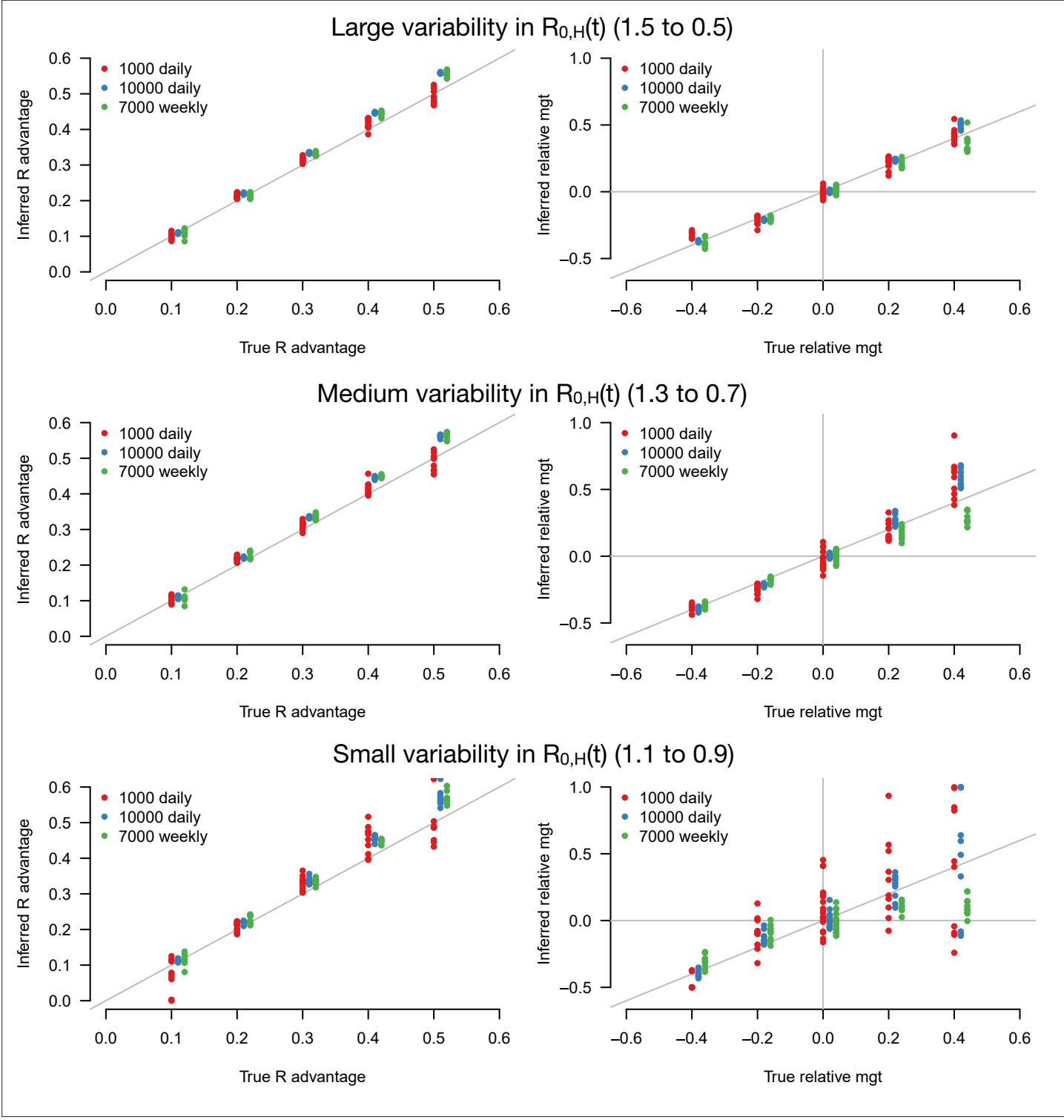

**Figure 3.** Inference of the R advantage and relative mean generation time in the simulation study. The plots show the correlation between inferred and true quantities. The three colors show different sample sizes used to infer variant frequency: 1000 daily, 10,000 daily, and 1000 daily grouped by week (7000 weekly). The left panels show the R advantage, the right panels the mean generation time (mgt). Top, middle, and bottom panels show inference for large, medium, and small variability in $R_{0,H}(t)$. The simulations are initialized with $I_H(0,0) = 4000$ and $I_E(0,0) = 80$. For the inference of the R advantage, we used a variant relative mean generation time of $\delta_2 = +0\%$ (also inferred by the model), while for the inference of relative mean generation time we used the following combinations: $\{\delta_1 = 0.3, \ \delta_2 = -0.4\}$; $\{\delta_1 = 0.3, \ \delta_2 = -0.2\}$; $\{\delta_1 = 0.3, \ \delta_2 = 0\}$ ; $\{\delta_1 = 0.45, \ \delta_2 = +0.2\}$ ; $\{\delta_1 = 0.3, \ \delta_2 = +0.4\}$ . We used a greater R advantage when the mean generation time is unchanged or longer to ensure the emerging variant

*Figure 3 continued on next page*

*Figure 3 continued*

reaches a significant frequency over the 80 days time period considered. For each parameter combination, 10 replicate simulations were drawn with the same parameters, hence the same deterministic epidemiological dynamics, but with different random error on data (Poisson error on cases number and binomial error on frequencies).

The online version of this article includes the following figure supplement(s) for figure 3:

**Figure supplement 1.** The daily basic reproduction number of the historical strains and variant ($R_{0,H}(t)$ and $R_{0,E}(t)$, identical thick lines), together with the effective reproduction number inferred from daily case and frequency data (thin lines), in the example simulation shown on *Figure 2B* for an emerging variant with +10% effective reproduction number and same distribution of generation time.

**Figure supplement 2.** Inference of the R advantage and relative mean generation time in two additional simulation studies: (i) where the lag between symptom onset and case detection is longer (mean 6days instead of 2.2days) shown as points ("longer lag"), (ii) where the distribution of generation time is different from the assumptions of the model shown as open squares ("different gtd").

**Figure supplement 3.** Inference of the R advantage and relative mean generation time (mgt) in the additional simulation study where the level of transmission is assumed to sharply decline from $R_{0,H}(t) = 1.5$ to $R_{0,H}(t) = 0.5$.

The variant Alpha swept through from 0% to almost 100% frequency over 25 weeks; data were of very good quality for the whole period (*Figure 4*). Using our framework, we found that Alpha had a R advantage equal to $\delta_1 = 54\%$ $[45, 63\%]$, and the same mean generation time as previous strains, $\delta_2 = 0.058$ $[-0.095, 0.21]$ (*Figure 5*).

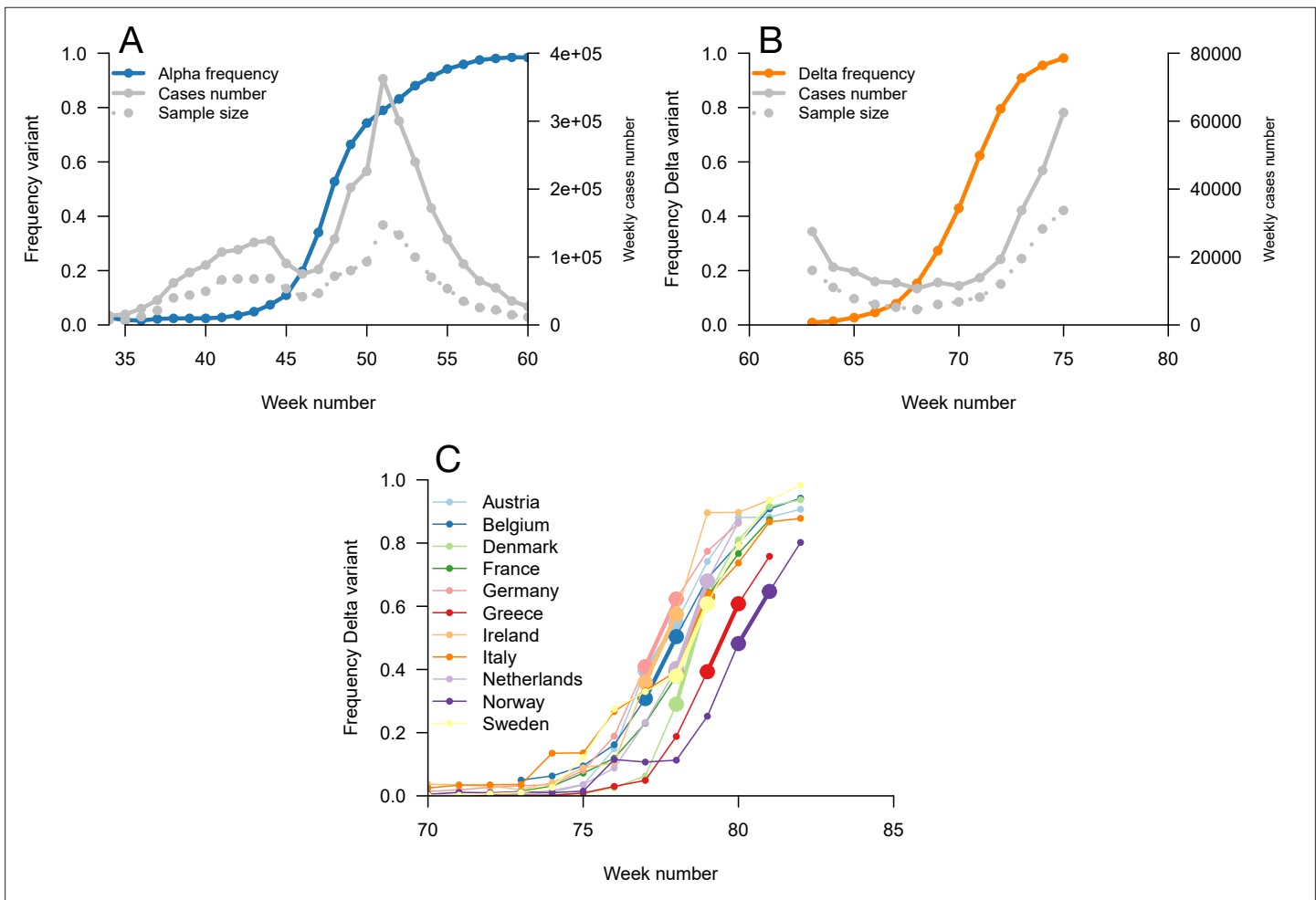

**Figure 4.** Data used for inference. A and B: Dynamics of the Alpha variant frequency in England (**A**) and of the Delta variant frequency in England (**B**), estimated through SGTF. These frequencies are shown together with the dynamics of weekly cases numbers, over weeks 35 (week starting September 08, 2020)–60 (week starting March 02, 2021) for Alpha, and weeks 63 (starting March 23, 2021)–75 (starting June 15, 2021) for Delta. (**C**) the dynamics of the Delta variant frequency in 11 chosen European countries. For each country, we used the growth rate of historical strains and emerging Delta variant when the frequency passes 50%, highlighted with larger points and thicker line on the figure.

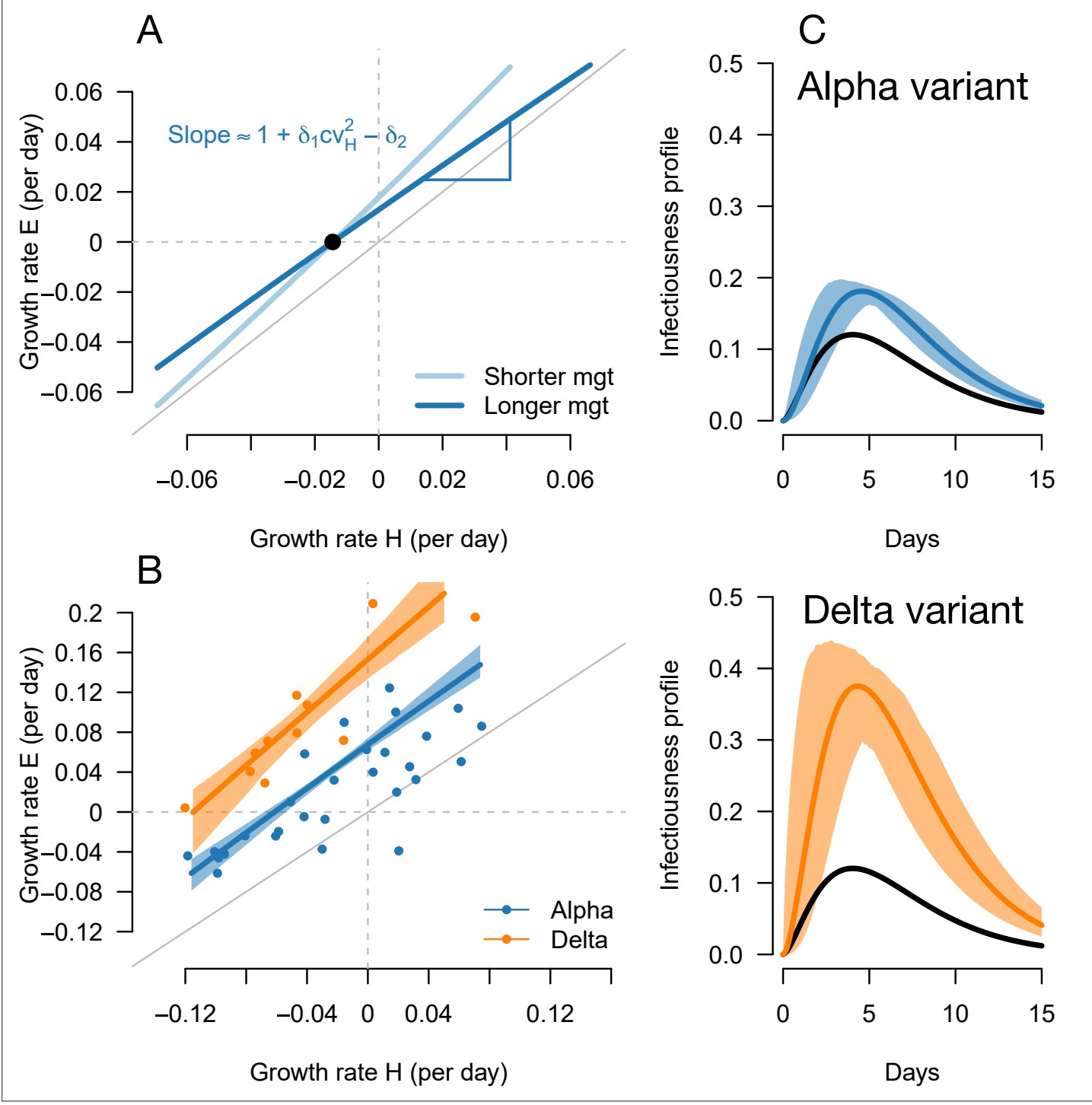

**Figure 5.** Inference of the infectivity profiles of the Alpha and Delta variants. (**A**) Geometric intuition for how the $(r_H, r_E)$ relationship depends on the mean generation time for hypothetical variants with R advantage $\delta_1 = 0.1$ and relative mean generation time of $\delta_2 = -0.15$ and $\delta_2 = +0.15$ (as in *Figures 1 and 2*). The parameter $cv_H$ denotes the coefficient of variation of the historical strains distribution of generation time, equal to the standard deviation (sd) divided by the mean of the distribution. (**B**) The correlation between estimated growth rates of historical strains and variants in England. For Alpha (dataset 1) this is a temporal correlation in England over weeks starting September 08, 2020 to March 16, 2021. For Delta (dataset 3),this is a *spatial* correlation across European countries. The curves with confidence intervals are the model fits for each variant and show that the selection coefficient (the vertical distance to the bisector) is roughly constant across epidemiological conditions, indicative that the mean generation time is not altered in these variants. (**C**) The inferred infectivity profile of each variant with confidence intervals, together with the infectivity profile of historical strains circulating before the rise of the Alpha variant (black curve). This inference assumes a gamma-distributed generation time.

*Figure 5 continued on next page*

*Figure 5 continued*

The online version of this article includes the following figure supplement(s) for figure 5:

**Figure supplement 1.** The relationship between the selection coefficient and level of transmission $R_H(t)$ for the Alpha variant spreading in England.

**Figure supplement 2.** The effective reproduction number of historical strains $R_H(t)$ estimated on the English data, for the period when the Alpha variant emerged and replaced historical strains, then the period when the Delta variant emerged and replaced the Alpha variant and other strains.

**Figure supplement 3.** Negative log-likelihood of the fully optimized model as a function of overdispersion in number of cases and in emerging variant frequency.

The information on the relative mean generation time of a variant mainly lies in the slope of the relationship between $r_E$ and $r_H$. In fact, the slope is approximately $1 + cv_H^2 \, \delta_1 - \delta_2$ for variants not too different from historical strains, where $cv_H$ is the coefficient of variation of the distribution of generation time of historical strains (Materials and methods). A variant with an R advantage but not changing the mean generation time ($\delta_1 > 0$, $\delta_2 = 0$) would present a slope slightly greater than 1. This geometric intuition is just a different way to interpret the increasing selection coefficient on variants conferring an R advantage as levels of transmission increase (*Figure 1C and D*, gray lines). A variant shortening the mean generation time would present a steeper slope. A variant prolonging the mean generation time would present a flatter slope (*Figure 5A*, *Figure 2—figure supplement 1*). For the Alpha variant, the decline in selection coefficient expected as levels of transmission declined in England is seen in our data as in previous studies (*Volz et al., 2021b*; *Otto et al., 2021*). It is apparent in *Figure 4* (the frequency trajectory slows down at around 70%) and we highlight it more clearly in *Figure 5—figure supplement 1*.

We next evaluated the growth of Delta (B.1.617.2) in England. As Delta does not have the spike 69/70 deletion, the growth of Delta and eventual replacement of Alpha in April and May 2021 was evidenced by the decline of SGTF (*Public Health England, 2021c*) (dataset 2). However, we found that with this temporal data, we could not reliably disentangle the R advantage from the mean generation time. The confidence intervals were very wide, with an estimated R advantage equal to $\delta_1 = 229\%$ $[7, 451\%]$ and a relative mean generation time estimated at $\delta_2 = 1$ $[-0.31, 2.3]$. This was linked to the small temporal variations in epidemic conditions in this short period of time compared to the previous period when we investigated Alpha (*Figure 5—figure supplement 2*).

In an attempt to gain more power to disentangle the R advantage from the mean generation time for the Delta variant, we used *spatial* variation in growth rates across European countries. We used data from the European Surveillance System (TESSy) on the growth rate of "historical" strains circulating at the time when Delta emerged (mainly Alpha variant), and the emerging Delta variant across 11 European countries with sufficient genomic data (Austria, Belgium, Denmark, France, Germany, Greece, Ireland, Italy, Netherlands, Norway, Sweden: dataset 3). With these data, we inferred a R advantage equal to $\delta_1 = 140\%$ $[98, 182\%]$ compared to Alpha, and a mean generation time similar to the Alpha variant with $\delta_2 = 0.033$ $[-0.18, 0.25]$.

In conclusion, we found that the Alpha variant had a R advantage of +54% compared to historical strains, and the Delta variant had a further R advantage of +140% relative to Alpha, assuming a mean generation time of 6.5 days. There was no evidence of an altered mean generation time for these two variants.

## Discussion

We investigated how emerging variants with distinct infectivity profiles may be selected. Our main finding is that levels of transmission reflected in the reproduction number *R(t)*, which depend on human behavior and interventions, change selection on different types of variants. It was known that selection on variants with an R advantage but not affecting the mean generation time is stronger when the transmission is high. We extend previous work by investigating selection on both R and the timing of secondary infections. In a context of high transmission and high growth rate ("fast" epidemic), most infections will be recent and it is more advantageous to transmit early. Conversely, in a context of low transmission ("slow" or declining epidemic), infections will be older and it is more advantageous to transmit late. The selection coefficient on variants may thus increase or decrease with the transmission, and it can even be a non-monotonous function of transmission. A second important finding is that the variation of the selection coefficient with transmission provides insight on both the

R advantage and the infectivity profile. We used this understanding to infer variant infectivity profiles from the variation in growth rates in contexts of changing transmission.

We found that Alpha variant enjoy a R advantage of +54% [45, 63] relative to historical strains, and Delta variants a+140% [98, 182] additional advantage relative to Alpha. Both variants have a mean generation time similar to that of historical strains (here assumed to be 6.5 days). This complements previous findings: the possibility of a shorter generation time was investigated to explain the growth advantage of Alpha, but previous studies found it was difficult to distinguish a variant with larger R from one with a shorter generation time (*Volz et al., 2021b*; *Davies et al., 2021*). Some (but not all) studies of within-host viral load dynamics suggested that individuals infected by variant Alpha could shed virus for a longer time, which may result in a longer generation time (*Kissler et al., 2021*; *Cosentino et al., 2022*; *Elie et al., 2021*). Conflicting results exist for Delta. A cluster of Delta variant infections in China suggested a smaller mean generation time (2.9 days) and more pre-symptomatic transmission for Delta than early SARS-CoV-2 strains, but without good control (*Zhang et al., 2021*); more controlled studies found on the contrary a similar generation time (*Pung et al., 2021*; *Ryu et al., 2021*). If the distribution of generation time is similar in Delta as we infer, the observed growth rates imply it is +140% more transmissible than Alpha, in line with the +70 to +200% estimated elsewhere (*Alizon et al., 2021*; *Blanquart et al., 2021*). If the generation time of Delta had instead been inferred 1 or 2 days shorter than the baseline (6.5 days), the observed growth rate would imply the estimated R advantage drops to +106 or +77%. The large R advantage of Delta with similar mean generation time makes it more difficult to control with interventions reducing transmission (such as social distancing) than if it had a smaller R advantage associated with a shorter mean generation time.

Our conceptual results may be useful to interpret the dynamics of variant frequency dynamics. Frequency dynamics are determined by the selection coefficient, which is the difference in growth rates between the variant and historical strains. Several papers have reported a varying selection coefficient across time (*Roquebert et al., 2021b*) or across countries (*Campbell et al., 2021*). One possible explanation for this variability is that the selection coefficient changes with transmission because of differences in generation time distributions, which may explain non-monotonous frequency trajectories (*Figure 2C*).

Evolution influences epidemiological dynamics: at a given level of transmission, stronger selection on a variant also implies a larger incidence of the variant and therefore a larger burden on healthcare systems (*Figure 2A*). However, we emphasize that measures to reduce transmission, although possibly increasing the *selection coefficient*, always result in reduced variant *absolute growth rate* and better control of the variant. The selection coefficient (*i.e.* the difference in growth rates) can increase if reducing transmission reduces the growth rate of historical strains more strongly than the growth rate of the variant (e.g. variant conferring longer mean generation time, *Figure 1D*). It remains true that a variant epidemic growth rate is an increasing function of transmission, and a variant is growing if and only if $R_E > 1$ (*Equation 3*).

A first limitation of our framework is that we only consider the impact of changing transmission on selection for variants. We do not consider the impact of interventions shortening the distribution of generation time such as isolation of positive cases and contact tracing, which also potentially change over time (*Kraemer et al., 2021*). These interventions would alter the selection coefficient differently, in particular would favor earlier transmission (shorter mean generation time). A second important factor that could change selection for variants and is not considered in our framework is vaccination. Vaccination reduces host susceptibility. If a variant partially escapes vaccine immunity, rapid vaccination of the population will change the selection coefficient of the variant over time. For example, if a temporal reduction in transmission is solely caused by the deployment of vaccines, the growth rate of historical strains will be strongly reduced as the vaccination campaign progresses, but the growth rate of the variant will be less reduced. This could flatten the relationship between $r_E$ and $r_H$. The effects of the R advantage, a different generation time and vaccine escape on changing selection could in principle be disentangled with data on vaccine efficacy and the vaccination status of the host population over time. These considerations could analogously apply to the comparison across countries with different levels of vaccination. This might affect the analyses of the infectivity profile of the Delta variant that possibly confers immune escape. A third limitation, linked to the comparison across countries, is that we assume that the variability in growth rates across countries is solely explained by changing transmission. The R advantage and distribution of generation time are assumed to be

the same in all countries. Relaxing this assumption would greatly impede the possibility of inference of variant infectivity profiles using data sampled across countries. Fourthly, we consider only one emerging variant. Our inference framework could readily be extended without additional technical complications to the frequency dynamics of several variants. Lastly, our framework requires precise measure of variant frequency and varying levels of transmission over the timespan considered. Infectivity profiles can sometimes be inferred with as little as a 10 days time series after a sharp decline in transmission (*Figure 3—figure supplement 3*).

In spite of these limitations, our simple framework makes minimal assumptions (exponential growth rate and stable age-of-infection structure) that proved robust when tested against a simulation model including more complicated features (build-up of immunity, isolation of positive cases, explicit epidemiological dynamics in the context of changing transmission). We additionally verified the robustness of our framework when the lag between infection and cases is different, the true generation time is not gamma-distributed, and the decline in the transmission is very sharp (*Figure 3—figure supplements 2 and 3*).

In conclusion, we developed a conceptual framework to study selection on variants modifying the transmission of an infectious pathogen. We decoupled selection on R (number of secondary infections) and selection on the distribution of generation time (timing of secondary infections). While selection on the number of secondary infections is stronger with increased transmission (e.g. contact rates) in the community, selection on the generation time varies with levels of transmission in the community. This can lead to non-monotonous variant frequency trajectories. The ensuing variation in the selection coefficient in contexts of changing transmission can be used to infer not only the R advantage of variants, but also the change in mean generation time. This inference method could be used in conjunction with other type of data, for example, data on variant viral load trajectories (*Cosentino et al., 2022*; *Roquebert et al., 2021a*), or comparison of secondary attack rates of variant vs. historical strains informing on the R advantage (*Public Health England, 2021a*). We find that the patterns of growth of the Alpha and Delta variants in England and Europe are compatible with mean generation times similar to historical strains. This work will help understand variant dynamics at a time when several variants of concern are circulating and new ones may evolve, and genomic and PCR-based surveillance programs allow fine monitoring of their dynamics.

## Materials and methods
### The Model
### Relationship Between Growth Rate and Effective Reproduction Number

We describe the exponential growth of an epidemic within a simple framework. We assume that transmission changes over time because of varying social distancing measures. We assume that changes are sufficiently slow compared to the mean generation time that the distribution of time since infection is always at equilibrium. It is known that when the number of infected individuals grows exponentially at a constant rate $r$, the distribution of time since infection equilibrates to an exponential distribution with parameter $r$ (*Wallinga and Lipsitch, 2007*). Here, we assume that the number of infected individuals grows exponentially with rate $r(t)$ and that the distribution of the time since infection is exponential with parameter $r(t)$ (i.e., the distribution equilibrates instantaneously when $r(t)$ changes). The unit of time is days. The timing of infections is defined by the probability density $w(\tau)$. Under these assumptions, the effective reproduction number $R(t)$ reflecting transmission and the growth rate $r(t)$ are linked through (*Wallinga and Lipsitch, 2007*):

$$r(t) = \int_0^\infty \underbrace{r(t)e^{-r(t)\tau}}_{\substack{\text{stable age} \\ \text{structure}}} \underbrace{w(\tau)R(t)}_{\substack{\text{new infections} \\ \text{from individuals} \\ \text{of age } \tau}} d\tau. \tag{1a}$$

The growth rate is the integral over all possible times since infection of the probability that an individual has this age (the exponential distribution), times the number of new infections produced by an individual this age, giving the relation (1 a). This equation simplifies to:

$$1 = R\left(t\right) \int_0^\infty e^{-r(t)\,\tau}\, w\left(\tau\right)\, d\tau \tag{1b}$$

For a gamma-distributed generation time (with shape and scale parameters $\alpha$ and $\beta$), this integral has an explicit solution:

$$R\left(t\right) = \left(1 + \beta\, r\left(t\right)\right)^\alpha \tag{2}$$

Conversely, the growth rate is:

$$r\left(t\right) = \left(R\left(t\right)^{\frac{1}{\alpha}} - 1\right) / \beta \tag{3}$$

The epidemic grows when $R\left(t\right) > 1$ and declines when $R\left(t\right) < 1$. Reparameterizing in terms of the mean and sd of the gamma distribution, $\mu = \alpha\,\beta$ and $\sigma = \sqrt{\alpha}\,\beta$, yields:

$$r\left(t\right) = \left(R\left(t\right)^{\sigma^2/\mu^2} - 1\right)\left(\mu/\sigma^2\right). \tag{4}$$

## Relationship Between Growth Rates of Historical Strains and Emerging Variant Across Epidemiological Conditions

This equation can separately describe the growth of historical strains and variants characterized by their own parameters: $R_H\left(t\right), \mu_H$ and $\sigma_H$ for historical strains, $R_E\left(t\right), \mu_E$, and $\sigma_E$ for the emerging variant. We may recast the variant parameters in terms of how they are altered compared to historical strains:

$$\begin{aligned} R_E\left(t\right) &= R_H\left(t\right)\left(1 + \delta_1\right), \\ \mu_E &= \mu_H\left(1 + \delta_2\right), \\ \sigma_E &= \sigma_H\left(1 + \delta_3\right). \end{aligned} \tag{5}$$

We assume that changes in behavior and in interventions cause temporal variability in the transmission, measured by the effective reproduction number of historical strains taken as the reference, $R_H\left(t\right)$, and only in this parameter. This assumption is valid for interventions that reduce transmission (social distancing, vaccines) but not for interventions that change the distribution of generation time (contact tracing, case isolation). Temporal variability in $R_H\left(t\right)$ will affect both $r_H\left(t\right)$ and $r_E\left(t\right)$ through the relationship (***Borges et al., 2021***):

$$r_H\left(t\right) = \left(R_H\left(t\right)^{\frac{\sigma_H^2}{\mu_H^2}} - 1\right)\left(\mu_H/\sigma_H^2\right) \tag{6}$$

$$r_E\left(t\right) = \left(R_E\left(t\right)^{\frac{\sigma_E^2}{\mu_E^2}} - 1\right)\left(\mu_E/\sigma_E^2\right) = \left(R_H\left(t\right)^{\frac{\sigma_E^2}{\mu_E^2}}\left(1 + \delta_1\right)^{\frac{\sigma_E^2}{\mu_E^2}} - 1\right)\left(\mu_E/\sigma_E^2\right)$$

The temporal variation in $R_H\left(t\right)$ thus defines a parametric relationship between $r_H\left(R_H\left(t\right)\right)$ and $r_E\left(R_H\left(t\right)\right)$ that can be used to infer $\delta_1$, $\delta_2$ and $\delta_3$. The variation of the selection coefficient with $R_H\left(t\right)$ is simply given by $r_E\left(t\right) - r_H\left(t\right)$.

## Approximation of the Parametric Relationship

To gain further intuition on the parametric relationship between $r_H\left(R_H\left(t\right)\right)$ and $r_E\left(R_H\left(t\right)\right)$ as $R_H\left(t\right)$ varies, we compute the tangent of the curve at the value $R_H\left(t\right) = 1$ (where $r_H$ is 0). The parameters of this tangent will of course be most informative when values of $R_H\left(t\right)$ are not too far from one. The intercept is:

$$intercept = r_E\left(1\right) = \left(\left(1 + \delta_1\right)^{\frac{\sigma_E^2}{\mu_E^2}} - 1\right)\left(\mu_E/\sigma_E^2\right) \tag{7}$$

Where $\mu_E$ and $\sigma_E^2$ can be reparameterized with **Equation 5**. This intercept is approximately $\delta_1/\mu_H$ for a variant of small effect ($\delta_1$, $\delta_2$, and $\delta_3$ are all small). The equation for the slope of the tangent is complicated, but again assuming small-effect variant it can be approximated as:

$$slope \approx 1 + \delta_1\,\mathrm{cv}_H^2 - \delta_2 \tag{8}$$

with the square coefficient of variation of the generation time distribution, $\mathrm{cv}_H^2 = \sigma_H^2/\mu_H^2$. Here it is assumed to be equal to $4^2/6.5^2 = 0.38$. Thus, the slope of the $(r_H, r_E)$ relationship is close to one, increased by the R advantage (weighted by $\mathrm{cv}_H^2$), decreased by the change in mean generation time, and unaffected to the first order by the difference in sd $\delta_3$. The selection coefficient is $r_E - r_H$, the vertical distance between the $(r_H, r_E)$ relationship and the bisector.

## Likelihood

We now formulate the likelihood for the estimated growth rates across time and across regions, denoted by $\hat{r}_H$ and $\hat{r}_E$. We derive an approximation for the distribution of $\hat{r}_H$ and $\hat{r}_E$ as a function of their true values $r_H$ and $r_E$ and the error variances, under the assumption that these errors are normally distributed and small. The historical strains and emerging variant growth rates are estimated from daily cases and variant frequency time series, by decomposing the cases into historical and emerging variant cases. We will formalize below how the estimated growth rates are affected by the error on variant frequency which depends on $1/N$, where $N$ is sample size, and on the error on cases number.

Between two consecutive days, denoted for simplicity days 0 and 1, the observed variables are the total number of cases denoted $\hat{I}_0$ and $\hat{I}_1$ and the frequency of variant $\hat{p}_0$ and $\hat{p}_1$. The time is now noted as an index for clarity. The observed total number of cases is:

$$\begin{aligned} \hat{I}_0 &= I_0\left(1 + \epsilon_{I,0}\right) \\ \hat{I}_1 &= I_1\left(1 + \epsilon_{I,1}\right) \end{aligned} \tag{9}$$

We assume that the number of cases is Poisson distributed. When the number of cases is large, the error denoted $\epsilon_{I,0}$ (for day 0) is approximately normally distributed with mean 0 and variance $1/I_0$ (and analogously for day 1). If we now assume that the number of cases is distributed as a negative binomial (over-dispersed compared to the Poisson): when the parameter $r$ (number of failures) of the negative binomial is large, the normal approximation is accurate by the central limit theorem and the variance of $\epsilon_{I,0}$ will be $1/\left[I_0\left(1-\pi\right)\right]$ where $\pi$ is the success probability of the negative binomial. Indeed, we have $V\left[\hat{I}_0\right] = I_0^2\,\mathrm{V}\left[\epsilon_{I,0}\right] = I_0/\left(1-\pi\right)$. The second equality comes from the fact that the variance of a negative binomial distribution is the mean (here $I_0$) divided by $1-\pi$. In other words, the cases number can be reduced to a smaller effective cases number to account for such overdispersion.

The observed logit-frequency of the variant is:

$$\begin{aligned} \hat{lp}_0 &= lp_0 + \epsilon_{lp,0} \\ \hat{lp}_1 &= lp_1 + \epsilon_{lp,1} \end{aligned} \tag{10}$$

where $lp$ is a shorthand for the logit of the frequency of the variant. In the data, the frequency of the variant was estimated by running a specific PCR on a fraction on all cases and is given by a binomial frequency distribution. If the sample size is large, the error on the logit frequency $\epsilon_{lp,0}$ is approximately normally distributed within mean 0 and variance $1/\left(N_0\,p_0\left(1-p_0\right)\right)$ where $N_0$ is the sample size at day 0 (and similarly for day 1, variance $1/\left(N_1\,p_1\left(1-p_1\right)\right)$). Indeed, for a binomial distribution of variant cases, the error on the frequency is approximately normal and has variance $p_0\left(1-p_0\right)/N_0$. By application of the delta method, the error on the logit frequency has variance $1/\left(N_0\,p_0\left(1-p_0\right)\right)$. Here again, it is possible to account for overdispersion compared to the binomial distribution. The beta-binomial distribution can be used to reflect such overdispersion, as it is a compound distribution emerging when the frequency parameter of the binomial is itself distributed according to a beta distribution. The beta-binomial distribution is close to a normal distribution when the two parameters of the underlying beta ($\alpha$ and $\beta$) are large. The variance of the frequency for a beta-binomial distribution of variant cases is inflated by a factor $\left(\alpha + \beta + N_0\right)/\left(\alpha + \beta + 1\right)$ compared to a binomial, while

the mean is $p_0 = \alpha/(\alpha + \beta)$. Thus, overdispersion can be modeled by reducing the sample size to a smaller effective number.

The observed growth rates are linked with the observed variables through the relations:

$$\begin{aligned} \hat{r}_{H,1} &= \log[\hat{I}_{H,1}/\hat{I}_{H,0}] = \log[\hat{I}_1(1-\hat{p}_1)/(\hat{I}_0(1-\hat{p}_0))] \\ \hat{r}_{E,1} &= \log[\hat{I}_{E,1}/\hat{I}_{E,0}] = \log[\hat{I}_1\hat{p}_1/(\hat{I}_0\hat{p}_0)] \end{aligned}$$

(11)

## Small Error Approximation

Replacing the observed number of cases and frequencies with the expressions above, and approximating with a first order Taylor series in the error terms, yields:

$$\hat{r}_{H,1} = \text{r}_{H,1} + \underbrace{\epsilon_{I,1} - \epsilon_{I,0}}_{\substack{\text{error on case} \\ \text{incidence}}} + \underbrace{p_0\epsilon_{lp,0} - p_1\epsilon_{lp,1}}_{\substack{\text{error on} \\ \text{frequency}}}$$

(12)

$$\hat{r}_{E,1} = r_{E,1} + \epsilon_{I,1} - \epsilon_{I,0} - (1-p_0)\;\epsilon_{lp,0} + (1-p_1)\;\epsilon_{lp,1}$$

The first two error terms are common to both growth rates and simply express the fact that both growth rates will be inflated if the number of cases is by chance larger at time 1 than at time 0 ($\epsilon_{I,1} > \epsilon_{I,0}$). The two last terms express the fact that estimated growth rates will be modified by the error on the estimation of the frequency. For example, the growth rate of the historical strain will be inflated if by chance the error term on frequency at day 0 is greater than that at day 1 ($p_1\epsilon_{lp,1} < p_0\;\epsilon_{lp,0}$). These terms are modulated by the true frequencies $p_0$ and $p_1$.

Under the assumption that the errors on the logit frequencies and number of cases are independent and that they are also independent between day 0 and 1, the distribution of $(\hat{r}_{H,1}, \hat{r}_{E,1})$ is bivariate normal. The mean vector is:

$(r_{H,1}, r_{E,1})$.

The variances (diagonal of the variance-covariance matrix) are:

$$V\left[\hat{r}_{H,1}\right] = V\left[\epsilon_{I,1}\right] + V\left[\epsilon_{I,0}\right] + p_0^2V\left[\epsilon_{lp,0}\right] + p_1^2 V\left[\epsilon_{lp,1}\right] = 1/I_1 + 1/I_0 + p_0/(N_0\;(1-p_0)) + p_1/(N_1\;(1-p_1))$$

$$V\left[\hat{r}_{E,1}\right] = 1/I_1 + 1/I_0 + (1-p_0)/(N_0\;p_0) + (1-p_1)/(N_1\;p_1).$$

The covariances are:

$$\text{cov}\left[\hat{r}_{H,1}, \hat{r}_{E,1}\right] = (1/I_1 + 1/I_0) - (1/N_0 + 1/N_1).$$

The covariance between the estimated growth rates of historical and emerging strains is expected to be negative, because only a fraction of cases are assayed for the presence of the variant ($N_0 < I_0$). Indeed, the first positive term ($1/I_1 + 1/I_0$) expresses the fact that positive covariance will be created by error in cases number (for example, larger cases number will translate into higher growth rate of both historical strains and variants). The second negative term expresses the fact that the error in estimation of variant frequency will impact $\hat{r}_{H,1}$ and $\hat{r}_{E,1}$ in opposite ways. This second negative effect will be dominant.

## Definition of the Likelihood
### Likelihood for Temporal Dynamics
We now formulate more generally the likelihood of observed growth rates at all days $t$, $\hat{r}_{H,t}, \hat{r}_{E,t}$ with $t \in [1, t_{max}]$. Across consecutive timepoints, successive estimates of $\hat{r}_{H,1}, \hat{r}_{E,1}$ will also be correlated because they share the same error terms. These cross-time covariances will be:

$$\begin{aligned} \text{cov}\left[\hat{r}_{H,t}, \hat{r}_{H,t+1}\right] &= \text{cov}\left[\epsilon_{I,t} - p_t\epsilon_{lp,t}, -\epsilon_{I,t} + p_t\;\epsilon_{lp,t}\right] \\ &= -V\left[\epsilon_{I,t}\right] - p_t^2V\left[\epsilon_{lp,t}\right] \\ &= -1/I_t - p_t/(N_t\;(1-p_t)) \end{aligned}$$

$$\text{cov}\left[\hat{r}_{E,t}, \hat{r}_{E,t+1}\right] = -1/I_t - \left(1 - p_t\right) / \left(N_t\, p_t\right)$$

$$\text{cov}\left[\hat{r}_{E,t}, \hat{r}_{H,t+1}\right] = -1/I_t + 1/N_t$$

$$\text{cov}\left[\hat{r}_{H,t}, \hat{r}_{E,t+1}\right] = -1/I_t + 1/N_t$$

Finally, the likelihood for the estimated growth rates of historical strain and mutant within a region at all days is given by the density of the multivariate normal distribution:

$$P\left(\hat{r}_{H,1}, \hat{r}_{E,1}, \ldots, \hat{r}_{H,t_{max}}, \hat{r}_{E,t_{max}} \mid R_{H,1}, \ldots, R_{H,t_{max}}, \delta_1, \delta_2, \delta_3\right)$$

$$= f_{\boldsymbol{m},\boldsymbol{\Sigma}}\left(\hat{r}_{H,1}, \hat{r}_{E,1}, \ldots, \hat{r}_{H,t_{max}}, \hat{r}_{E,t_{max}}\right) \tag{13}$$

where $f_{\boldsymbol{m},\boldsymbol{\Sigma}}$ is the density of the multivariate normal distribution. The mean vector $\boldsymbol{m}$ is composed of the true growth rates, which depend in turn on the temporal series of true R of historical strains $R_{H,1}, \ldots, R_{H,t_{max}}$, the temporal series of true R of the variant $R_{H,1}\left(1 + \delta_1\right), \ldots, R_{H,t_{max}}\left(1 + \delta_1\right)$, and the distribution of the generation time of the variant characterized by $\delta_2, \delta_3$. The distribution of the generation time of historical strains is assumed to be known. The covariance matrix $\boldsymbol{\Sigma}$ is block-diagonal with non-zero covariances between mutant and historical strains at the same time and at consecutive time-points, as given above. The true number of cases $I_t$, sample size $N_t$, and the true variant frequencies $p_t$, intervene in the expressions for the variances and covariances. In practice, we approximate the true number of cases and variant frequencies by their estimations, $\hat{I}_t$ and $\hat{p}_t$.

## Likelihood for Spatial Variation

The likelihood of a series of growth rates $\hat{r}_{H,i}, \hat{r}_{E,i}$ collected in different regions or countries indexed by $i$ is different from that for a temporal series of $\hat{r}_{H,t}, \hat{r}_{E,t}$, as we assume there are no correlations between growth rates measured in different countries (in contrast to temporal correlations). For such data representing spatial variation in growth rates, the mean vector is composed of the true growth rates, which depend on the series of R in different countries, $R_{H,i}$. The covariance matrix has diagonal given by the variances:

$$V\left[\hat{r}_{H,i}\right] = 1/I_{1,i} + 1/I_{0,i} + p_{0,i}/\left(N_{0,i}\left(1 - p_{0,i}\right)\right) + p_{1,i}/\left(N_{1,i}\left(1 - p_{1,i}\right)\right)$$

$$V\left[\hat{r}_{E,i}\right] = 1/I_{1,i} + 1/I_{0,i} + \left(1 - p_{0,i}\right)/\left(N_{0,i}\, p_{0,i}\right) + \left(1 - p_{1,i}\right)/\left(N_{1,i}\, p_{1,i}\right).$$

Where 0 and 1 are the consecutive timepoints at which the growth rates are measured, and the index $i$ denotes the country. Furthermore, the covariances between growth rates of historical strains and emerging variants are:

$$\text{cov}\left[\hat{r}_{H,i}, \hat{r}_{E,i}\right] = 1/I_{1,i} + 1/I_{0,i} - 1/N_{0,i} - 1/N_{1,i}.$$

All other covariances (between quantities measured in different countries) are 0.

## Simulation Study

Our approach relies on several approximations. First, we assume that the age-of-infection structure of the population is always at equilibrium with the current growth rate $r\left(t\right)$ even though transmission $R\left(t\right)$ changes over time. Second, we assume that errors around cases number and frequencies are small and normally distributed. We conducted a simulation study to test our ability to infer the parameters of the variant infectivity profile in spite of these approximations.

## Description of the Simulation Model

The simulation model is as in a previous study (*Belloir and Blanquart, 2021*) and includes time-varying transmission, arbitrary infectivity profiles, the build-up of population immunity, and detection and isolation of cases. To model transmission dynamics, we use a discretized version of the renewal equation (see also *Flaxman et al., 2020*) extended to two strains, historical strains and the variant. We follow the dynamics of the number of individuals infected at day $t$ who were infected $\tau$ days ago, and have not yet been detected and isolated, called $I_H\left(t, \tau\right)$ and $I_E\left(t, \tau\right)$. The transmission dynamics are given by the system of recurrence equations:

$$I_H\left(t+1,\,0\right) = R_{0,H}(t)\ \underbrace{\left(1-I_t^{tot}/S_0\right)}_{\substack{\text{non immune}\\\text{fraction}}}\ \underbrace{\Sigma_{\tau=0}^{\infty}w_H(\tau)I_H(t,\tau)}_{\substack{\text{infectivity}\\\text{profile}}} \tag{14}$$

$$I_E\left(t+1,\,0\right) = R_{0,E}\left(t\right)\ \left(1-I_t^{tot}/S_0\right)\ \sum_{\tau=0}^{\infty}w_E\left(\tau\right)\ I_E\left(t,\,\tau\right)$$

$$I_H\left(t+1,\,\tau\right) = I_H(t,\tau-1)\ \underbrace{\left(1-cy(\tau-1)\right)}_{\substack{\text{case detection}\\\text{and isolation}}}\ \forall\tau\geq 1$$

$$I_E\left(t+1,\,\tau\right) = I_E(t,\tau-1)(1-cy(\tau-1))\forall\tau\geq 1$$

The first two equations are analogous and represent transmission to new susceptible individuals giving rise to infected individuals with time since infection 0. The parameter $R_{0,H}\left(t\right)$ reflects the transmission of historical strains, and is the basic reproduction number (in the absence of interventions, and when the population is fully susceptible (i.e. $I_t^{tot} = 0$)). The factor $w_H\left(\tau\right)$ is the fraction of transmission that occurs at time since infection $\tau$, where $\sum_{\tau=0}^{\infty}w_H\left(\tau\right) = 1$. Thus $w_H\left(\tau\right)$ represents the distribution of the generation time of the virus. In practice, we use a discretized version of the gamma distribution. The infectivity profile of the virus is the product of R and the generation time distribution $R_{0,H}\left(t\right)\ w_H\left(\tau\right)$. Transmission is reduced by a factor $1-I_t^{tot}/S_0$ by population immunity, where $S_0$ is the initial number of susceptible individuals in the region. Population immunity is assumed to be the same for both variants (perfect cross-protection). The variable $I_t^{tot} = \sum_{i=1}^{t}\left(I_H\left(i,0\right)+I_E\left(i,0\right)\right)$ is the total number of individuals already infected and assumed to be fully immune at time $t$. The instantaneous reproduction number that accounts for population immunity but not for case isolation is $R_H\left(t\right) = R_{0,H}\left(t\right)\ \left(1-I_t^{tot}/S_0\right)$.

The third and fourth equations are analogous and represent the dynamics of individuals infected in the past. Individuals infected $\tau-1$ days ago now have time since infection $\tau$, provided they were not detected and isolated. An infected individual is detected with probability $c$, and the probability that an individual is detected at age $\tau$ (when it is detected) is given by $y\left(\tau\right)$, with $\sum_{\tau=1}^{\infty}y\left(\tau\right) = 1$. The distribution $y\left(\tau\right)$ represents the lag between infection and case detection. An individual who is detected is removed from the pool of individuals that contribute to further transmission of the disease. The number of cases detected at day $t$ is thus:

$$C_H\left(t\right) = c\sum_{\tau=0}^{\infty}y\left(\tau\right)I_H\left(t,\tau\right) \tag{15}$$

$$C_E\left(t\right) = c\sum_{\tau=0}^{\infty}y\left(\tau\right)I_E\left(t,\tau\right)$$

And the number of detected individuals who were infected $\tau$ days ago changes as:

$$C(t+1,0) = 0 \text{ (when } \tau = 0) \tag{16}$$

$$C(t+1,\tau) = C(t,\tau-1) + c\,I(t,\tau-1)\,y(\tau-1)\ \forall\ \tau\geq 1$$

## Parameterization of the Simulation Model

We simulate the epidemiological dynamics of historical strains and variant virus. We assume that historical strains initially grow slowly, while the emerging variant initially increases in incidence faster thanks to its R advantage. Transmission progressively declines. This could be due for example to the progressive strengthening of control measures, leading to control of historical strains and then the variant. Over the period considered, the variant initially increases in frequency.

We ran the simulation for 80 days. We first ran a simulation where we assumed the basic reproduction number $R_{0,H}\left(t\right)$ changed as a Brownian motion with mean 1.5 and autocorrelation 0.05, and selected a trajectory such that the mean $R_{0,t}$ over the 20 first days exceeded 1.1 and the mean over the 20 last days was below 0.9 (**Figure 2**). For systematic simulations designed to test the inference algorithm, we chose three scenarios for the temporal variation in $R_{0,H}\left(t\right)$: linear decline from 1.5 to 0.5, from 1.3 to 0.7, and from 1.1 to 0.9. We assumed about 9 M individuals are initially susceptible (corresponding to a large European region), and an initial number of detected cases of 4000 historical strains infections per day and 80 variant infections per day. We assumed 50% of all infection are detected (probability of detection $c = 0.5$). The time to detection has mean 7.3 days. It is the sum of

time from infection to symptom onset with mean 5.1 days (distributed as log-normal with parameters 1.518, 0.472 [*Lauer et al., 2020*]), and the time from symptom to detection with mean 2.2 days (distributed as gamma with shape 0.69, rate 0.31 [*Belloir and Blanquart, 2021*; *Lauer et al., 2020*]). For the alternative simulations with a longer lag (mean 6 days), we assumed a gamma distribution with shape 0.69, rate 0.115. This ensures a mean of 6 days and the same coefficient of variation as for the main parameter set.

We assumed the generation time of historical strains was gamma distributed with mean 6.5 days and sd 4 days (*Volz et al., 2021b*). For the alternative simulations where we assume that there is zero transmission the first two days.

## Inference

For inference, we tested combinations of the $\delta_1$, $\delta_2$ parameters describing how the variant differs from the historical strains in its R and mean generation time. In the terms of the simulation model parameters, $R_{0,E}(t) = R_{0,H}(t)(1 + \delta_1)$. The parameter $\delta_2$ affects the parameters of the gamma distribution describing $w_E(\tau)$, while $w_H(\tau)$ is gamma-distributed with mean 6.5days and sd 4days. We assumed the variant had the same sd of generation time, $\delta_3 = 0$. Indeed, variants affecting the sd of generation time distribution have a very small selection coefficient which makes inference of this parameter difficult. This is seen by initial exploration of the relationship between growth rates of historical strains and variant, $\hat{r}_H$ and $\hat{r}_E$ (*Figure 2—figure supplement 1*).

We systematically tested our ability to infer the $\delta_1$ and $\delta_2$ parameters. We ran the simulation model for all combinations of $\delta_1$ from 0 to 0.5 in steps of 0.1 (R advantage increased from +0 to +100%), and of $\delta_2$ from –0.4 to 0.4 in steps of 0.2. We inferred the parameters $\delta_1$ and $\delta_2$ by maximum likelihood using the expression for the likelihood in *Equation 13*.

### Heuristic for the Simulation Study

In principle we need to infer jointly the parameters of interest $\{\delta_1, \delta_2\}$, together with the other unknown parameters $\{R_{H,1}, \ldots, R_{H,t_{max}}\}$, which can be many. We used a simplified heuristic whereby we first set the parameters $\{R_{H,1}, \ldots, R_{H,t_{max}}\}$ to plausible values given $\{\delta_1, \delta_2\}$, then infer the maximum likelihood parameters $\{\delta_1, \delta_2\}$ given $\{R_{H,1}, \ldots, R_{H,t_{max}}\}$, and so on. We inferred maximum likelihood parameters $\{\delta_1, \delta_2\}$ using the "BFGS" (Broyden, Fletcher, Goldfarb, and Shanno) method implemented in the optim function in the software R (*R Development Core Team, 2018*). In the simulation study, we iterate this procedure five times with five different starting points for $\{\delta_1, \delta_2\}$ and select the final maximum likelihood parameter set $\{\delta_1, \delta_2, R_{H,1}, \ldots, R_{H,t_{max}}\}$.

To set $\{R_{H,1}, \ldots, R_{H,t_{max}}\}$ to plausible values, we used the measured growth rates with a simplified likelihood function. The simplified likelihood does not consider the full covariance structure of the multivariate normal distribution but instead only use the variances of the distribution $V[\hat{r}_{H,i}]$ and $V[\hat{r}_{E,i}]$ expressed above. Given $\{\delta_1, \delta_2\}$, we set the historical strains transmissibilities to:

$$R_{H,i} = \left[ \frac{1}{V[\hat{r}_{H,i}]} \left(1 + \frac{\sigma^2}{\mu} \hat{r}_{H,i}\right)^{\mu^2/\sigma^2} + \frac{1}{V[\hat{r}_{E,i}]} \frac{1}{1+\delta_1} \left(1 + \frac{\sigma^2}{\mu(1+\delta_2)} \hat{r}_{E,i}\right)^{\mu^2(1+\delta_2)^2/\sigma^2} \right] \left(\frac{1}{V[\hat{r}_{H,i}]} + \frac{1}{V[\hat{r}_{E,i}]}\right)^{-1} \quad (17)$$

This equation stems from *Equation 3* linking growth rates with effective reproduction number. It is reparameterized in terms of μ and $\sigma$, the mean and sd of the distribution of the generation time of the historical strains. It is a weighted sum of the effective reproduction number as estimated from the historical strains growth rate $\hat{r}_{H,i}$, and the effective reproduction number as estimated from the variant growth rate $\hat{r}_{E,i}$. For the latter, the mean generation time is altered by $(1 + \delta_2)$, and the R is altered by $(1 + \delta_1)$. The weights are the inverse variance of each of the estimated growth rates, $1/V[\hat{r}_{H,i}]$ and $1/V[\hat{r}_{E,i}]$.

The estimated growth rate of the historical strains and variants at time $i$ are given by *Equation 11* applied to each time-point:

$$\hat{r}_{H,i} = \log[\hat{I}_{i+1}(1 - \hat{p}_{i+1})/(\hat{I}_i(1 - \hat{p}_i))]$$
$$\hat{r}_{E,i} = \log[(\hat{I}_{i+1} \hat{p}_{i+1}) / (\hat{I}_i \hat{p}_i)] \quad (18)$$

We systematically verified in the simulations that this heuristic converged to a set of $R_{H,i}$ close to the true effective reproduction number.

## Full Inference for Analysis of English and European Data

We ran inference on three datasets, two based on temporal variation in growth rates in England, and the third based on spatial variation across European countries:

1. Data on the growth of the Alpha variant in England from September 8, 2020 to March 16, 2021. The frequency of the Alpha variant is estimated from "S-gene target failures" which reflect the deletion 69–70 in the gene S typical of Alpha.
2. Data on the growth of the Delta variant in England from March 23, 2021 to June 15, 2021. The frequency of the Delta variant is again estimated from "S-gene target failures," where this time the "historical strains" is the Alpha variant and the Delta variant which does not have the deletion 69–70 in gene S emerges.
3. TESSy data from the European Centre for Disease Prevention and Control (*European Centre for Disease Prevention and Control, 2021*), on the growth of the Delta variant based on viral isolates sequenced in 11 European countries: Austria, Belgium, Denmark, France, Germany, Greece, Ireland, Italy, Netherlands, Norway, Sweden. We selected countries with sufficient data (based on visual inspection of the frequency trajectory of Delta), and used for each country the growth rates between the two weeks when the frequency of the Delta variant passed 50%.

As these datasets present a limited number of weeks or countries, it was possible to directly infer jointly the complete parameter set, $\{\delta_1, \delta_2, R_{H,1}, \ldots, R_{H,t_{max}}\}$ for temporal datasets, $\{\delta_1, \delta_2, R_{H,1}, \ldots, R_{H,n_c}\}$ for the spatial dataset where $n_c = 11$ is the number of countries. Furthermore, we ran optimizations for several effective number of cases and sample sizes, to select the best amount of overdispersion (*Figure 5—figure supplement 2*). Each optimization is conducted with 30 iterations of the BFGS algorithm with 30 random initial parameter values, and selecting the final best optimization. The 95% confidence intervals on the parameters were computed assuming multivariate normality of the likelihood function, and estimating the Hessian matrix of this multivariate normal at the optimum. All codes are shared on GitHub (*Blanquart, 2022*; copy archived at swh:1:rev:a242a18349393e2e98d353a879ef9e66e99f21c1).

# Acknowledgements

FB was funded by a Momentum grant from CNRS. We thank Public Health England and The European Surveillance System (TESSy) of the European Centre for Disease Prevention and Control for sharing data on variant frequency.

# Additional information

### Funding

| Funder | Grant reference number | Author |
| --- | --- | --- |
| Centre National de la Recherche Scientifique | Momentum to FB | François Blanquart |

The funders had no role in study design, data collection and interpretation, or the decision to submit the work for publication.

### Author contributions

François Blanquart, Conceptualization, Data curation, Formal analysis, Investigation, Methodology, Visualization, Writing - original draft; Nathanaël Hozé, Conceptualization, Writing - review and editing; Benjamin John Cowling, Writing - review and editing; Florence Débarre, Simon Cauchemez, Conceptualization, Methodology, Writing - review and editing

### Author ORCIDs

François Blanquart ⓘ http://orcid.org/0000-0003-0591-2466
Benjamin John Cowling ⓘ http://orcid.org/0000-0002-6297-7154

Florence Débarre (iD) http://orcid.org/0000-0003-2497-833X

**Decision letter and Author response**
Decision letter https://doi.org/10.7554/eLife.75791.sa1
Author response https://doi.org/10.7554/eLife.75791.sa2

---

## Additional files

### Supplementary files
• Transparent reporting form

### Data availability
The codes used for the analyses are available on GitHub (copy archived at swh:1:rev:a242a18349393e2e-98d353a879ef9e66e99f21c1) along with the data used for the analysis (previously available data that is publicly available). A cleaned version of the data used for the analysis is also available on GitHub.

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
