## [Editor Report]

This manuscript will be of broad interest to readers interested in understanding the characteristics of variants in ongoing epidemics that lead to faster (or slower) growth rates and will be of particular interest to those wishing to understand the factors leading to the selection of SARS–CoV–2 variants. The selective advantage of a novel strain of a pathogen depends not only on its relative transmissibility but also on its generation time relative to other strains; the relation between transmissibility, transmission advantage and generation time changes across different phases of the epidemic. Key innovations in this paper are a robust framework for using this relationship to make statistical inferences about both the transmissibility advantage and generation time of an emerging variant and conceptual novelty in the general investigation of selection on infectivity profiles. The approach is supported by simulation studies and applied to the Alpha and Delta SARS–CoV–2 variants to show that selection was likely driven by changes in transmissibility rather than changes in the generation time.

---

## [Decision Letter]

**Decision letter after peer review:**

Thank you for submitting your article "Selection for infectivity profiles in slow and fast epidemics, and the rise of SARS–CoV–2 variants" for consideration by *eLife*. Your article has been reviewed by 2 peer reviewers, and the evaluation has been overseen by a Reviewing Editor and Miles Davenport as the Senior Editor. The reviewers have opted to remain anonymous.

Essential revisions:

1. There is a need to clarify the relationship between changes in transmission and changes in selection more explicitly in the text. (specific suggestions for how this could be done are given by reviewer #1)

2. Relevant previous work should be cited and the abstract should more clearly highlight the novelty of this work

3. Additional analysis to establish robustness of the conclusions to assumptions would be helpful (as outlined by reviewer #1)

4. Please highlight the broader applicability. As currently framed the emphasis is on changes in transmissibility due to control measures, but similar considerations apply if there are changes over time in transmissibility for other reasons.

5. Emphasise limitations, for example caveats related to the possible immune escape properties of the Delta variant , and potential biases introduced by vaccination.

6. Could the authors comment on the amount of necessary data (length of the time series) for the method to be useful

7. Clarify multiple country Delta analysis to address reviewer #2's concern.

*Reviewer #1 (Recommendations for the authors):*

That said, I found myself confused about several points. Most of these issues can be easily clarified, and I encourage the authors to tackle these points to avoid confusion.

The first confusion stems from the idea, described in Day et al. (2020), that selection for transmission isn't constant but declines when contact rates decline. Subsequently, during this pandemic, we have repeatedly observed a decrease in selection for VOC when public health and individual NPI measures increase, as seen as a decrease in s with stringency index (Figure 3, Otto et al. 2021) and as a "kink" in frequency–over–time plots (Figure S2, Otto et al. 2021). This kink is also visible in the current manuscript (Figure 4A), with Α being more strongly favoured prior to week ~50 than after.

It was easy as a reader, however, to think that selection on transmission was modelled in this paper as a constant, not varying with control measures. Diving into the methods more, this isn't the case, but it was easy to miss it. I would suggest clarifying the relationship between changes in transmission and changes in selection more explicitly in the text.

For example:

– Equation (5) – I suggest renaming the terms s1–s3 as delta1–delta3 or some other letter. Labeled with an "s", like the selection coefficient makes it easy to think that selection for transmission is s1. In fact, selection for transmission is given by rE–rH, as in equation (6), which is not linear in the s1–s3 terms. Even when R~1 (r~0), selection is s1/muH. This is because selection to leading order depends on changes in the transmission rate (call this β), not changes in R, consistent with equations (1) and (S6) in Day et al. [This is visible in the grey curves in Figure 1C,D, but I think most people would miss this.]

– In a related way, I saw Figure 5 and thought the analysis was assuming that selection was constant as rH varies. This isn't assumed but is a function of R being near enough to one over the x–axis that the curves do look linear. Still, if one plotted the selection (s=rE–rH) versus rH, it would be easier to see that selection does depend on the growth rate in the model, weakening as rH decreases, even over this span of rH values.

– What about the kink? This analysis hides a bit the temporal changes in selection. Figure 5B makes it seem like selection is constant over the time course, but Figure 4A (and logit plots) shows more of a breakpoint, with stronger selection earlier and weaker selection later. I would suggest some discussion of this; while the kink looks discrete in estimates of selection from Figure 4A, the interpretation in this paper would be that the kink is caused by a sharp reduction in rH, but this is hard to see in Figure 5B (potentially because of noise in the estimates of rH and rE). Any larger change in selection, beyond that predicted by changes in rH, could be examined by fitting a breakpoint in the likelihood analysis (future work). I think that this could be brought out more clearly as a strength of the paper – we expect the strength of selection to change with changing control measures, but the current work attempts to estimate the constant changes to the underlying parameters.

– Because R describes the transmission per generation, wouldn't inferences of s1 capture both changes in transmissibility and changes in the generation time (mu)? However, s1 is discussed only in terms of a change in transmissibility. Wouldn't it be clearer to dissect the two (R=β–(mean generation time)) and focus on selection via the transmission rate per unit time (β) versus selection on the mean generation time? Alternatively, care should be taken throughout to avoid talking about s1 as a change in transmissibility (it is more accurately described on line 358 as a change in the number of secondary infections).

In addition to clarifying the above, other issues that I think would strengthen the paper are:

Citing previous work, not just Day et al. but also the earlier literature (e.g., refs 11, 13, 14, 19 therein). The abstract and text (e.g., line 115, 300) makes it seem like this is the first work to dissect selection into components coming from different disease life history traits (transmission, time to infectiousness, clearance rate, etc.), but there is substantial prior work on this. The novelty, and the focus of the abstract, should be on inference of these components.

It would be good to get a better sense of the robustness of the inferences to changing various assumptions:

– Non–gamma infectiousness curves (e.g., a curve that adds a delay to the start of infectiousness).

– Lags in measuring case case numbers, as individuals typically don’t get tested until having symptoms. A lag is in the simulation, but I didn’t get a sense of how changing the lag affects the inferences.

– Allowing shifts in contact rates or NPI – the method assumes steady state distributions at all times, it might be easier to get at the transients by having a discrete change in r from high to low values and seeing how the inference of the s terms change over time.

*Reviewer #2 (Recommendations for the authors):*

– I think the emphasis on "level of epidemic control" is misplaced. The whole analysis may work also in an uncontrolled epidemic, right? All that matters is that the effective reproduction number varies significantly with time.

– I would discuss briefly the biases that could arise from incomplete cross–protection or different vaccine protection, in order to clarify the applicability of the method. At present, I think the limitations are not clear enough, and they should be emphasised.

– This approach seems to require a quite dense/long time series. Could the authors comment on this requirement?

– I find the notation for the parameters s1,s2,s3 most unintuitive: s1 is a selection coefficient, but the others aren't, and s2 actually behaves in the opposite way. I'd suggest to pick a more intuitive notation, e.g. R_rel_,gt_rel_,sd_rel_ or any other variant.

– For the Δ analysis: could you briefly discussed the caveats related to the possible immune escape properties of the variant?

– Also for the Δ analysis: multiple countries can differ in their epidemiological situation and therefore correspond to different combinations of R and generation time for the Α variant. It is unclear to me how this may affect your cross–country analysis.

---

## [Author Response]

Essential revisions:1. There is a need to clarify the relationship between changes in transmission and changes in selection more explicitly in the text. (specific suggestions for how this could be done are given by reviewer #1)

We followed the reviewers’ advice and clarified the relationship between changes in transmission and changes in selection at lines 139 (and also lines 333, 404):

“We first remind that even for variants with an R advantage but not affecting the distribution of generation time, the selection coefficient increases with transmission (9–11). This relationship is shown on Figure 1 as grey lines for a series of variant conferring a R advantage of +10% to +50% but not affecting the gamma-distributed generation time.”

We also added Figure 5-figure supplement 1 showing these changes for the case of the Α variant.

2. Relevant previous work should be cited and the abstract should more clearly highlight the novelty of this work.

We now describe and cite the suggested additional literature lines 76-89 and modified the abstract to more clearly highlight the novelty of this work:

“It is known that selection on a variant with larger R increases with levels of transmission in the community. We additionally show that variants conferring earlier transmission are more strongly favoured when the historical strains have fast epidemic growth, while variants conferring later transmission are more strongly favoured when historical strains have slow or negative growth”.

3. Additional analysis to establish robustness of the conclusions to assumptions would be helpful (as outlined by reviewer #1).

We have done three additional analyses to further test the robustness of the conclusions to assumptions.

– New simulations where the lag between symptom onset and case detection has a mean of 6 days instead of 2.2 days as assumed in the mean set of simulations. This does not change the inference. This is expected as long as cases and variant frequencies share the same lag between infections and case detection, as argued lines 298-303:

“First, we investigated whether temporal lag between infections and recorded cases impacted inference, considering the additional scenario where the mean time from symptom onset to case detection was 6 days instead of 2.2 days (Figure 3 —figure supplement 2). The lag does not affect the accuracy of inference (compare with Figure 3), as expected since the time series of number of detected cases and variant frequency share the same lag with infections.”

– New simulations where the distribution of the generation time is not gamma-distributed, but instead has density 0 up to day 2 then is distributed as a shifted gamma distribution (line 303).

“Second, we applied our framework to additional simulations where the true generation time is not gamma-distributed as in our baseline model: we assumed that transmission is not possible during the first two days; this is followed by a shifted gamma-distributed timing of secondary infections. This mismatch between modelling assumption and the true generation time distribution led to a small overestimation of the relative mean generation time (Figure 3 —figure supplement 2).”

– New simulations where transmission declines abruptly. We investigate the accuracy of parameter inference as a function of the number of days for which the number of cases and variant frequency data are available after the sharp decline from 10 to 30 days (line 308).

“Lastly, we inferred parameters when there is a sharp decline in R_H_(t) from 1.5 to 0.5 at a fixed date, instead of a progressive decline. In this scenario, the distribution of time since infection may not immediately stabilise after the sharp decline, temporarily breaking the key assumption of our analytical approach. Parameters were correctly inferred when the time series after the decline was long enough. With only 10 days of data after the decline, the inferred relative mean generation time was accurately inferred if enough cases were assayed to measure the variant frequency (Figure 3 —figure supplement 3). Interestingly, accuracy was better for variants with a shorter generation time than the historical strains (negative relative mean generation time). This is probably because their distribution of time since infection stabilises faster after the sharp decline.”

4. Please highlight the broader applicability. As currently framed the emphasis is on changes in transmissibility due to control measures, but similar considerations apply if there are changes over time in transmissibility for other reasons.

We wrote lines 95-97 that changes in transmission can occur for different reasons. We removed the emphasis on control measures throughout.

5. Emphasise limitations, for example caveats related to the possible immune escape properties of the Delta varian , and potential biases introduced by vaccination.

We developed the discussion of this limitation lines 459-473:

“A second important factor that could change selection for variants and is not considered in our framework is vaccination. Vaccination reduces host susceptibility. If a variant partially escapes vaccine immunity, rapid vaccination of the population will change the selection coefficient of the variant over time. For example, if a temporal reduction in transmission is solely caused by the deployment of vaccines, the growth rate of historical strains will be strongly reduced as the vaccination campaign progresses, but the growth rate of the variant will be less reduced. This could flatten the relationship between r_E_ and r_H_. The effects of the R advantage, a different generation time and vaccine escape on changing selection could in principle be disentangled with data on vaccine efficacy and the vaccination status of the host population over time. These considerations could analogously apply to the comparison across countries with different levels of vaccination. This might potentially affect the analyses of the infectivity profile of the Delta variant that possibly confers immune escape.”

6. Could the authors comment on the amount of necessary data (length of the time series) for the method to be useful

We comment on the amount of necessary data in the discussion lines 480-483:

“Lastly, our framework requires precise measure of variant frequency and varying levels of transmission over the timespan considered. Infectivity profiles can sometimes be inferred with as little as a 10-days time series after a sharp decline in transmission (Figure 3 —figure supplement 3).”

7. Clarify multiple country Delta analysis to address reviewer #2's concern.

We added some caveats associated with the analysis of multiple countries lines 468-477:

“These considerations could analogously apply to the comparison across countries with different levels of vaccination. This might potentially affect the analyses of the infectivity profile of the Delta variant that possibly confers immune escape. A third limitation, linked to the comparison across countries, is that we assume that the variability in growth rates across countries is solely explained by changing transmission. The R advantage and distribution of generation time are assumed to be the same in all countries. Relaxing this assumption would greatly impede the possibility of inference of variant infectivity profiles using data sampled across countries.”

Reviewer #1 (Recommendations for the authors):That said, I found myself confused about several points. Most of these issues can be easily clarified, and I encourage the authors to tackle these points to avoid confusion.The first confusion stems from the idea, described in Day et al. (2020), that selection for transmission isn't constant but declines when contact rates decline. Subsequently, during this pandemic, we have repeatedly observed a decrease in selection for VOC when public health and individual NPI measures increase, as seen as a decrease in s with stringency index (Figure 3, Otto et al. 2021) and as a "kink" in frequency–over–time plots (Figure S2, Otto et al. 2021). This kink is also visible in the current manuscript (Figure 4A), with Α being more strongly favoured prior to week ~50 than after.It was easy as a reader, however, to think that selection on transmission was modelled in this paper as a constant, not varying with control measures. Diving into the methods more, this isn't the case, but it was easy to miss it. I would suggest clarifying the relationship between changes in transmission and changes in selection more explicitly in the text.

We fully agree with this comment and clarified this upfront line 139:

“We first remind that even for variants with an R advantage but not affecting the distribution of generation time, the selection coefficient increases with transmission (9–11). This relationship is shown on Figure 1 as grey lines for a series of variant conferring a R advantage of +10% to +50% but not affecting the gamma-distributed generation time.”

For example:– Equation (5) – I suggest renaming the terms s1–s3 as delta1–delta3 or some other letter. Labeled with an "s", like the selection coefficient makes it easy to think that selection for transmission is s1. In fact, selection for transmission is given by rE–rH, as in equation (6), which is not linear in the s1–s3 terms. Even when R~1 (r~0), selection is s1/muH. This is because selection to leading order depends on changes in the transmission rate (call this β), not changes in R, consistent with equations (1) and (S6) in Day et al. [This is visible in the grey curves in Figure 1C,D, but I think most people would miss this.]

We followed this suggestion and renamed the parameters δ_1_, δ_2_, δ_3_.

– In a related way, I saw Figure 5 and thought the analysis was assuming that selection was constant as rH varies. This isn't assumed but is a function of R being near enough to one over the x–axis that the curves do look linear. Still, if one plotted the selection (s=rE–rH) versus rH, it would be easier to see that selection does depend on the growth rate in the model, weakening as rH decreases, even over this span of rH values.

Thank you. We agree. We plotted the selection coefficient as a function of time, and the selection coefficient as a function of transmission, on Figure 5 —figure supplement 1 shown above. This shows more clearly the decline in selection as transmission declines in England.

We refer to this new figure in the text at lines 330-342:

“The information on the relative mean generation time of a variant mainly lies in the slope of the relationship between r_E_ and r_H_. In fact, the slope is approximately 1+ cvH2δ1−δ2 for variants not too different from historical strains, where cv_H_ is the coefficient of variation of the distribution of generation time of historical strains (Methods). A variant with an R advantage but not changing the mean generation time (δ_1_ > 0, δ_2_ = 0) would present a slope slightly greater than 1. This geometric intuition is just a different way to interpret the increasing selection coefficient on variants conferring an R advantage as levels of transmission increase (Figure 1C and D, gray lines). A variant shortening the mean generation time would present a steeper slope. A variant prolonging the mean generation time would present a flatter slope (Figure 5A, Figure 2 —figure supplement 1). For the Α variant, the decline in selection coefficient expected as levels of transmission declined in England is seen in our data as in previous studies (2,16). It is apparent on Figure 4 (the frequency trajectory slows down at around 70%) and we highlight it more clearly on Figure 5 —figure supplement 1.”

– What about the kink? This analysis hides a bit the temporal changes in selection. Figure 5B makes it seem like selection is constant over the time course, but Figure 4A (and logit plots) shows more of a breakpoint, with stronger selection earlier and weaker selection later. I would suggest some discussion of this; while the kink looks discrete in estimates of selection from Figure 4A, the interpretation in this paper would be that the kink is caused by a sharp reduction in rH, but this is hard to see in Figure 5B (potentially because of noise in the estimates of rH and rE). Any larger change in selection, beyond that predicted by changes in rH, could be examined by fitting a breakpoint in the likelihood analysis (future work). I think that this could be brought out more clearly as a strength of the paper – we expect the strength of selection to change with changing control measures, but the current work attempts to estimate the constant changes to the underlying parameters.

We discuss the “kink” in the frequency of variant Alpha over time at lines 330-342, and show it more explicitly on Figure 5 —figure supplement 1. We do not attempt to fit two coefficients of selection but a breakpoint is indeed visible on Figure 5 —figure supplement 1B around week 49.

– Because R describes the transmission per generation, wouldn't inferences of s1 capture both changes in transmissibility and changes in the generation time (mu)? However, s1 is discussed only in terms of a change in transmissibility. Wouldn't it be clearer to dissect the two (R=β–(mean generation time)) and focus on selection via the transmission rate per unit time (β) versus selection on the mean generation time? Alternatively, care should be taken throughout to avoid talking about s1 as a change in transmissibility (it is more accurately described on line 358 as a change in the number of secondary infections).

We agree that “transmissibility” wrongly suggests a change in rate of transmission (per unit time) while it is a change in effective reproduction number—the total number of secondary infections produced by one infection. We have replaced throughout with “effective reproduction number”, and “transmissibility advantage” by “R advantage”.

In addition to clarifying the above, other issues that I think would strengthen the paper are:Citing previous work, not just Day et al. but also the earlier literature (e.g., refs 11, 13, 14, 19 therein). The abstract and text (e.g., line 115, 300) makes it seem like this is the first work to dissect selection into components coming from different disease life history traits (transmission, time to infectiousness, clearance rate, etc.), but there is substantial prior work on this. The novelty, and the focus of the abstract, should be on inference of these components.

Thank you. We have added citation to the suggested literature. We agree there is substantial prior work on selection acting on different pathogen life history traits. We also agree the inference method we propose is novel. Moreover, we think there is conceptual novelty in that we investigate selection on infectivity profiles more generally than previous work. This is enabled methodologically by the use of the Lotka-Euler equation to study selection on R and the distribution of the generation time. Most previous work focused on the trade-off between transmission and virulence (disease-induced mortality), not on the distribution of generation time. There is some discussion of selection on the time between infection and infectiousness in Day et al. Current Biology 2020, but no systematic examination of the magnitude and sign of this component in relation to levels of transmission. We have amended the abstract and introduction to highlight more precisely the novelty of our work compared to previous work, most prominently at lines 76-89:

“Previous theoretical work has explored how selection acts on various pathogen life-history traits (8– 11), with most often a focus on selection for transmissibility and virulence (disease-induced mortality). Recent work on SARS-CoV-2 mentioned the possibility of selection on the time interval between infection and infectiousness, but did not characterise the sign and magnitude of this component of selection in details (11).”

It would be good to get a better sense of the robustness of the inferences to changing various assumptions:– Non–gamma infectiousness curves (e.g., a curve that adds a delay to the start of infectiousness).

Done with new simulations assuming zero transmission for the first two days, then a shifted gamma-distributed timing of secondary infections past day two. This does not alter inference much (line 303, Figure 3 —figure supplement 2).

– Lags in measuring case case numbers, as individuals typically don’t get tested until having symptoms. A lag is in the simulation, but I didn’t get a sense of how changing the lag affects the inferences.

We now discuss the lag at line 298. We ran additional simulations with a longer lag to check this does not affect our results (Figure 3 —figure supplement 2). The inference rests on cases number and frequencies, two time series that have the same lag with respect to the time of infections. Thus, the inference does not require considerations of this lag. The lag only affects the time difference between transmission in the community and the R(t) inferred from daily cases data.

– Allowing shifts in contact rates or NPI – the method assumes steady state distributions at all times, it might be easier to get at the transients by having a discrete change in r from high to low values and seeing how the inference of the s terms change over time.

We now conduct the suggested simulation to see how the inferred parameters change over time after an abrupt decline in transmission (line 308, Figure 3 —figure supplement 3). Inference is very poor if there is little data after the decline. Interestingly the inference of mean generation time is better for short generation time variants for which presumably the time-since-infection structure stabilizes to its equilibrium faster after the sharp decline.

Reviewer #2 (Recommendations for the authors):– I think the emphasis on "level of epidemic control" is misplaced. The whole analysis may work also in an uncontrolled epidemic, right? All that matters is that the effective reproduction number varies significantly with time.

Indeed, we removed the emphasis on level of epidemic control and now write more generally about the level of transmission as measured by the effective reproduction number.

– I would discuss briefly the biases that could arise from incomplete cross–protection or different vaccine protection, in order to clarify the applicability of the method. At present, I think the limitations are not clear enough, and they should be emphasised.

We emphasise this limitation in the Discussion lines 459-473:

“A second important factor that could change selection for variants and is not considered in our framework is vaccination. Vaccination reduces host susceptibility. If a variant partially escapes vaccine immunity, rapid vaccination of the population will change the selection coefficient of the variant over time. For example, if a temporal reduction in transmission is solely caused by the deployment of vaccines, the growth rate of historical strains will be strongly reduced as the vaccination campaign progresses, but the growth rate of the variant will be less reduced. This could flatten the relationship between r_E_ and r_H_. The effects of the R advantage, a different generation time and vaccine escape on changing selection could in principle be disentangled with data on vaccine efficacy and the vaccination status of the host population over time. These considerations could analogously apply to the comparison across countries with different levels of vaccination. This might potentially affect the analyses of the infectivity profile of the Delta variant that possibly confers immune escape.”

– This approach seems to require a quite dense/long time series. Could the authors comment on this requirement?

We emphasise this requirement in the discussion lines 480-483. We believe the time series needs to cover different levels of transmission and that the variant frequency needs to be precisely measured. But the time series need not be very long as shown by the additional simulations: a shorter mean generation time can be inferred with a time series extending 10 days after the sharp decline in transmission:

“Lastly, our framework requires precise measure of variant frequency and varying levels of transmission over the timespan considered. Infectivity profiles can sometimes be inferred with as little as a 10-days time series after a sharp decline in transmission (Figure 3 —figure supplement 3).”

– I find the notation for the parameters s1,s2,s3 most unintuitive: s1 is a selection coefficient, but the others aren't, and s2 actually behaves in the opposite way. I'd suggest to pick a more intuitive notation, e.g. R_rel_,gt_rel_,sd_rel_ or any other variant.

We agree and now followed the suggestion of reviewer 1 to name these parameters δ_1_, δ_2_, δ_3_.

– For the Delta analysis: could you briefly discussed the caveats related to the possible immune escape properties of the variant?

We now mention this caveat line 472 in relation to the caveats related to variability in immunity over time or across countries.

“This might potentially affect the analyses of the infectivity profile of the Delta variant that possibly confers immune escape.”

– Also for the Delta analysis: multiple countries can differ in their epidemiological situation and therefore, correspond to different combinations of R and generation time for the Α variant. It is unclear to me how this may affect your cross–country analysis.

We now discuss how the variability across countries in both the R and the generation time of a variant could affect the cross-country analysis lines 475-477:

“The R advantage and distribution of generation time are assumed to be the same in all countries. Relaxing this assumption would greatly impede the possibility of inference of variant infectivity profiles using data sampled across countries.”